# QAC: Quantization-Aware Conversion for Mixed-Timestep Spiking Neural Networks

## Abstract

Spiking Neural Networks (SNNs) have recently garnered widespread attention due to their high computational efficiency and low energy consumption, possessing significant potential for further research. Currently, SNN algorithms are primarily categorized into two types: one involves the direct training of SNNs using surrogate gradients, and the other is based on the mathematical equivalence between ANNs and SNNs for conversion. However, both methods overlook the exploration of mixed-timestep SNNs, where different layers in the network operate with different timesteps. This is because surrogate gradient methods struggle to compute gradients related to timestep, while ANN-to-SNN conversions typically use fixed timesteps, limiting the potential performance improvements of SNNs. In this paper, we propose a Quantization-Aware Conversion (QAC) algorithm that reveals a profound theoretical insight: the power of the quantization bit-width in ANN activations is equivalent to the timesteps in SNNs with soft reset. This finding uncovers the intrinsic nature of SNNs, demonstrating that they act as activation quantizers—transforming multi-bit activation features into single-bit activations distributed over multiple timesteps. Based on this insight, we propose a mixed-precision quantization-based conversion algorithm from ANNs to mixed-timestep SNNs, which significantly reduces the number of timesteps required during inference and improves accuracy. Additionally, we introduce a calibration method for initial membrane potential and thresholds. Experimental results on CIFAR-10, CIFAR-100, and ImageNet demonstrate that our method significantly outperforms previous approaches.

## 1 INTRODUCTION

Spiking Neural Networks (SNNs), as the third generation of neural networks, are inspired by the way biological neurons transmit information through spikes(Maass, 1997). Neurons in SNNs communicate using sparse and discrete spikes, with complex membrane potential updates and neural dynamic processes (Izhikevich, 2003; Ghosh-Dastidar & Adeli, 2009), which gives SNNs higher biological plausibility compared to traditional Artificial Neural Networks (ANNs). Recently, SNNs have garnered widespread attention due to their energy-efficient computational paradigm. It is well known that the human brain operates at approximately 20W of power while containing around 86 billion neurons(Herculano-Houzel, 2009), enabling it to perform complex reasoning and decision-making tasks. In contrast, current state-of-the-art artificial intelligence models require vast computational resources(Brown, 2020; Patterson et al., 2021; Xu & Poo, 2023), including hundreds of server racks and thousands of GPUs to support inference, consuming substantial amounts of energy. This significant energy consumption has raised widespread concerns due to its high economic cost. As a result, researchers are turning to SNNs, relying on their energy-efficient computing paradigm to reduce computational energy costs.

There are fundamental differences in the computational mechanisms between SNNs and ANNs (Pfeiffer & Pfeil, 2018). The computation in ANNs can be simplified as performing linear transformations within each layer, while continuous real-valued information is passed between layers via differentiable nonlinear activation functions. In SNNs, the results of linear transformations accumulate in the neuron's membrane potential, and spikes are triggered by non-differentiable nonlinear activation functions (Stöckl & Maass, 2021). The information passed between layers is in the form of discrete spike signals. Due to the advantage of sparse spike-based computation, recent studies

have focused on fully leveraging the energy-efficient characteristics of SNNs to achieve low-power computational models.

Currently, SNN learning methods are mainly categorized into two approaches: The first is through surrogate gradient methods, which replace the non-differentiable activation functions of SNNs with differentiable functions, enabling efficient training via backpropagation(Lee et al., 2016; Neftci et al., 2019; Wu et al., 2018b; Lee et al., 2020; Deng et al., 2022; Li et al., 2021b). The second approach exploits the mathematical equivalence between ANNs and SNNs, where the firing rates of SNN spikes approximate the activations of ANNs, allowing pre-trained ANNs to be converted into SNNs(Cao et al., 2015; Diehl et al., 2015; Rueckauer et al., 2016; Sengupta et al., 2019; Tavanaei et al., 2019; Kim et al., 2020; Li et al., 2021a; Bu et al., 2023).

Our work focuses primarily on the ANN-to-SNN conversion algorithm, aiming to achieve low-timestep, low-power, and high computational efficiency inference by converting pre-trained ANNs into SNNs. Specifically, our contributions include the following:

- We propose Quantization-Aware Conversion (QAC), an ANN-to-SNN algorithm based on mixed precision quantization, which can obtain SNNs with mixed time steps and high accuracy. In addition, we revealed the equivalence between the quantization bit-width of ANN activations and the timesteps in SNNs with soft-threshold resets.

- We found that when the weights are fixed, the residual membrane potential is related to the initial membrane potential and the threshold. We introduced a method for calibrating the initial membrane potential and the threshold to further improve the model's accuracy.

- The results on the CIFAR-10, CIFAR-100, and ImageNet datasets demonstrate that, compared to previous conversion methods, our approach achieves higher accuracy with fewer time steps. For instance, ResNet18 achieves 95.29 % accuracy on CIFAR-10 using only 2.76 time steps.

## 2 RELATED WORKS

**Mixed Precision Quantization.** Early quantization approaches typically applied the same quantization bit-width across different layers (Zhang et al., 2018; Choi et al., 2018; Bhalgat et al., 2020). However, different layers in neural networks exhibit varying sensitivities to quantization. When constrained by a uniform average bit-width, using the same quantization bit-width across all layers can lead to performance degradation (Dong et al., 2019). To address this issue, mixed-precision quantization methods assign different bit-widths to different parts of the model to balance "performance-efficiency": layers more sensitive to quantization are allocated higher bit-widths to minimize performance loss due to quantization, while layers less sensitive to quantization are assigned lower bit-widths to reduce storage and computational demands. The current mixed-precision quantization approaches can be categorized into four types: (1) Optimization based on sensitivity metrics: HAWQ (Dong et al., 2020; Yao et al., 2021) were among the first proposed methods for mixed bit-width quantization. These methods use loss and Hessian matrix information of model weights to gauge the quantization sensitivity of each layer and select quantization bit-widths accordingly. (2) Optimization using reinforcement learning: Methods like ReLeQ Elthakeb et al. (2020) and HAQ (Wang et al., 2019) employ reinforcement learning to allocate mixed bit-widths. Their state space includes different quantization bit-widths, and the reward is either the ratio of quantized accuracy to floating-point accuracy or a combination of task performance and simulated hardware performance. (3) NAS-based solutions: DNAS (Wu et al., 2018a) draws on differentiable search works DARTS(Liu et al., 2018), utilizing gradient-based methods to optimize bit-widths. Subsequent efforts like HMQ (Habi et al., 2020) fall under this category. (4) Learning-based mixed precision quantization solutions: (Uhlich et al., 2019; Wang et al., 2020; Yang & Jin, 2021)

**ANN-to-SNN Conversion.** The initial studies on ANN-to-SNN conversion were undertaken by (Cao et al., 2015), (Diehl et al., 2015; Rueckauer et al., 2016; Sengupta et al., 2019) further narrowed the gap between ANNs and SNNs through scaling and normalization of weights. Han & Roy (2020) proposed the use of soft-reset spiking neurons to further reduce conversion errors and minimize information loss. (Deng & Gu, 2021; Li et al., 2021a)revealed conversion errors by categorizing them into clipping and quantization error. (Ho & Chang, 2021) introduced Trainable Clipping Layers (TCL) to set thresholds effectively. (Bu et al., 2023) building on (Li et al., 2021a) work,

introduced "unevenness error", further refining the error analysis theory for ANN-to-SNN conversion. To further improve the accuracy of the converted SNN, (Wang et al., 2022) proposed signed spiking neurons model to enhance neuron performance. (Li et al., 2021a) introduced quantization fine-tuning to calibrate weights and biases, adjusting the biases at each layer under the assumption of a uniform current distribution. (Bu et al., 2022) assumed a uniform distribution of activations and demonstrated that half of the threshold is the optimal initial membrane potential. By adjusting the initial membrane potential to this value, neurons could spike more uniformly. However, as noted by (Datta & Beerel, 2022), the assumption of uniform activation distribution is incorrect, and thus a more detailed distribution function was used to optimize the activation distribution. (Hao et al., 2023a;b) proposed an optimization strategy based on Residual Membrane Potential (SRP), which effectively reduces "unevenness error" at low latency. Given the exceptional performance of transformer model architectures, some studies (Wang et al., 2023; You et al., 2024; Jiang et al., 2024b) have considered converting transformer structures.

## 3 PRELIMINARIES

### 3.1 NEURON MODEL

In Spiking Neural Networks (SNNs), the soft-reset Integrate-and-Fire (IF) neuron model (Cao et al., 2015) is commonly used. A key feature of this model is the soft-reset mechanism (Han et al., 2020), where the membrane potential is updated instead of being reset to a fixed value. The membrane potential is given by:

$$\boldsymbol{v}^l(t) = \boldsymbol{m}^l(t) - \boldsymbol{v}_{th}^l \boldsymbol{s}^l(t) \tag{1}$$

where $\boldsymbol{v}^l(t)$ represents the membrane potential of neurons in the $l$-th layer after a spike at time $t$, while $\boldsymbol{v}_{th}^l$ is the firing threshold, and $\boldsymbol{s}^l(t)$ denotes the spike output at time $t$. The membrane potential before the spike firing is as follows:

$$\boldsymbol{m}^l(t) = \boldsymbol{v}^l(t-1) + \boldsymbol{W}^l \boldsymbol{v}_{th}^{l-1} \boldsymbol{s}^{l-1}(t) \tag{2}$$

where $\boldsymbol{m}^l(t)$ denotes the postsynaptic membrane potential accumulated from the previous time step $\boldsymbol{x}^l(t-1)$ and synaptic input at the current time step of layer $l$. The neuron fires a spike when its membrane potential $\boldsymbol{m}^l(t)$ exceeds the threshold $\boldsymbol{v}_{th}^l$. The spike firing function is typically defined using the Heaviside function $H(\cdot)$, as follows.

$$\boldsymbol{s}^l(t) = H(\boldsymbol{m}^l(t) - \boldsymbol{v}_{th}^l) \tag{3}$$

### 3.2 CONVERSION FRAMEWORK

We follow the general conversion rules outlined in (Bu et al., 2022; 2023), transferring the weights from ANNs to SNNs. The forward propagation Equation 4 for SNNs is derived by substituting Equation 2 into Equation 4, summing both sides, and then leveraging the discrete nature of spike generation in SNNs. The detailed derivation of Equation 1 can be found in the Appendix.

$$\Phi^l(T_l) = \frac{\boldsymbol{v}_{th}^l}{T_l} \text{clip} \left( \left\lfloor \frac{\boldsymbol{W}^l \Phi^{l-1}(T_l) \cdot T_l - \boldsymbol{v}^l(T_l) + \boldsymbol{v}^l(0)}{\boldsymbol{v}_{th}^l} \right\rceil, 0, T_l \right) \tag{4}$$

where, $T_l$ represents the timestep of the $l$-th layer, and $\boldsymbol{v}_{th}^l$ denotes the threshold, $\lfloor \cdot \rceil$ denotes round function. $\Phi^l(T_l)$ is the equivalent output of the $l$-th layer in the SNN, $\Phi^l(T_l) = \frac{\sum_{t=1}^{T_l} \boldsymbol{s}^l(t) \boldsymbol{v}_{th}^l}{T_l}$. The general framework for converting ANNs to SNNs aims to approximate the equivalent output $\Phi^l(T_l)$ of each SNN layer to the corresponding ANN output $x^l$, i.e. $\boldsymbol{x}^l \approx \Phi^l(T_l)$. When $\boldsymbol{x}^{l-1} = \Phi^{l-1}(T_{l-1})$ and the residual term $\epsilon = \frac{\boldsymbol{v}^l(T_l) - \boldsymbol{v}^l(0)}{\boldsymbol{v}_{th}^l}$ is sufficiently small, the equivalent conversion from ANN to SNN can be achieved, i.e. $\boldsymbol{x}^l = \Phi^l(T_l)$. For the purpose of simplifying the derivation of the formulas, Bu et al. (2022; 2023); Li et al. (2021a) assume that the residual membrane potenti $\boldsymbol{v}^l(T) \in [0, \boldsymbol{v}_{th}^l]$. To improve conversion accuracy, (Hao et al., 2023a) calibrates the remaining membrane potential. (Li et al., 2021a) assumes the initial membrane potential $\boldsymbol{v}^l(0) = 0$, while (Bu et al., 2023) retains the initial membrane potential $\boldsymbol{v}^l(0)$ and sets $\boldsymbol{v}^l(0) = \frac{\boldsymbol{v}_{th}^l}{2}$ during SNN inference. We do not make assumptions about retaining $\epsilon$ and calibrate the initial membrane potential $\boldsymbol{v}^l(0)$ and the remaining membrane potential $\boldsymbol{v}_{th}^l$ in Section 4.3.

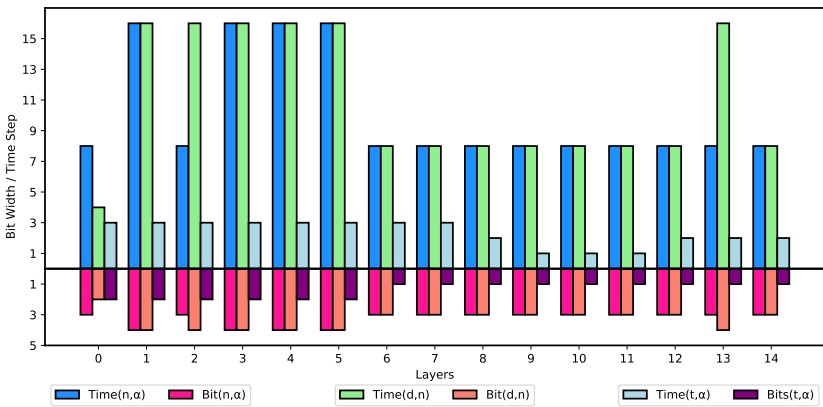

Figure 1: The quantization bit-width of each layer in VGG-16 for ANNs and the corresponding timesteps for SNNs on the CIFAR-10 dataset.

## 4 METHORDS

In this section, we first establish the connection between the quantization bit-width of ANNs and the timesteps in SNNs. Building on this insight, we propose a mixed-timestep conversion method for SNNs. Additionally, we discover that SNNs are highly sensitive to the initialization of membrane potential. To further enhance the accuracy of SNNs, we introduce a calibration method for the initial membrane potential and threshold.

### 4.1 THE EQUIVALENCE RELATIONSHIP BETWEEN ANN AND SNN

Quantization maps $\boldsymbol{x} \in \mathbb{R}$ to discrete values $\{q_1, ..., q_I\}$ through two steps: clipping and projection. Clipping constrains the values within a specific range from $q_{min}$ to $q_{max}$, while projection maps them to predefined quantization levels. When ReLU activations are quantized into unsigned integers, where $q_{min} = 0$, $q_{max} = 2^n - 1$, and $v_{max} = d \cdot q_{max}$, where $v_{max}$ is the maximum value of the activation. To further improve the performance of quantizers, several works (Choi et al., 2018; Gong et al., 2019; Bhalgat et al., 2020)have treated quantization parameters as learnable variables, updating them through gradients during Quantization-Aware Training (QAT). When $v_{max}$ is a learnable parameter $\alpha$, and the activation quantization levels are not strictly constrained to unsigned integer quantization $2^n - 1$, in the context of SNN conversion instead we use $2^n$, the activation quantization formula can be expressed as follows, where $\lfloor \cdot \rceil$ denotes the round function:

$$Q(\boldsymbol{x}) = \frac{\alpha}{2^n} \cdot \text{clip}\left(\left\lfloor \boldsymbol{x} \cdot \frac{2^n}{\alpha} \right\rceil ; 0, 2^n\right) \tag{5}$$

This is consistent with (Bu et al., 2023) approach to minimizing conversion error by using the quantization clip-floor-shift (QCFS) function to train quantized ANNs. QCFS utilize shared parameters within layers, represented as $\lambda$. For simplicity, we use the parameter $\alpha$ to represent the shared parameter. The learnable clipping threshold $\alpha$ can adaptively adjust the size of the quantization step for the activation output.

We further reveal the equivalence between the quantization levels of ANN activation outputs and the timesteps in the converted SNNs, leading to the following theorem.

**Theorem 1.** The timesteps of SNNs with soft reset are equivalent to the quantization levels of activations in ANNs.

$$T_{SNN} \simeq 2^n_{ANN} \tag{6}$$

*Proof.* Equation 4 represents the inference process of SNNs, where $\Phi^{l-1}(T)$ denotes the output of the $(l-1)$-th layer in SNNs. Equation 5 can be further expressed as: $\boldsymbol{x}^l = \frac{\alpha}{2^n} \cdot$ clip $\left(\left\lfloor \frac{\boldsymbol{W}^l \boldsymbol{x}^{l-1} \cdot 2^n}{\alpha} \right\rceil ; 0, 2^n\right)$, where $\boldsymbol{x}^l$ and $\boldsymbol{W}^l$ represent the output and weights of the $l$-th layer,

respectively. By comparing equations 4 and 6, we observe that when $\boldsymbol{x}^{l-1} = \Phi^{l-1}(T)$ and the residual term $\epsilon = \frac{\boldsymbol{v}^l(T_l) - \boldsymbol{v}^l(0)}{\boldsymbol{v}_{th}^l}$ is sufficiently small, the equivalent conversion from ANN to SNN can be achieved. Moreover, the timesteps $T$ in SNNs are equivalent to the quantization levels $2^n$ in ANNs.

Theorem 1 further explains why increasing inference timesteps in SNNs enhances performance—because adding timesteps is analogous to increasing the activation quantization bit-width in ANNs. Some early ANN-to-SNN conversion methods (Cao et al., 2015; Diehl et al., 2015; Rueckauer et al., 2016; Sengupta et al., 2019) typically required hundreds of timesteps. However, Theorem 1 indicates that many of these timesteps are redundant, given that low-bit quantized ANNs already exhibit high precision Zhou et al. (2016); Jacob et al. (2018); Bai et al. (2020); Kim et al. (2022); Li et al. (2022), only $2^n$ timesteps are required to achieve comparable accuracy to the ANN, where $n$ is typically a small value, typically around 2 or 3. Additionally, in ANNs, $n$-bit activations are multiplied by $m$-bit weights during forward inference. When converted to SNNs, this process simplifies to summing $2^n$ $m$-bit weights, which significantly reduces the memory and computational costs associated with activations, particularly in hardware implementations.

Table 1: Comparison between our method and previous works on CIFAR-10 dataset.

| Architecture | Methods | ANN(%) | Time Steps | SNN(%) |
|---|---|---|---|---|
| VGG-16 | RMP (Han et al., 2020) | 93.63 | 64 | 90.35 |
| | TSC (Han & Roy, 2020) | 93.63 | 64 | 92.79 |
| | RTS (Deng & Gu, 2021) | 95.72 | 64 | 90.64 |
| | SNNC-AP(Li et al., 2021a) | 95.72 | 32 | 93.71 |
| | SNM (Wang et al., 2022) | 94.09 | 32 | 93.43 |
| | OPI(Bu et al., 2022) | 94.57 | 8, 16 | 90.96, 93.38 |
| | QCFS(Bu et al., 2023) | 95.52 | 4, 8 | 93.96, 94.95 |
| | SlipReLU(Jiang et al., 2023) | 93.02 | 4, 8 | 91.08, 92.26 |
| | SGDND(Oh & Lee, 2024) | 95.96 | 16, 32 | 81.06, 95.53 |
| | **ours** | 95.12 | **2.33** | **94.78** |
| ResNet-18 | RMP (Han et al., 2020) | 91.47 | 128 | 87.60 |
| | TSC (Han & Roy, 2020) | 91.47 | 128 | 88.57 |
| | RTS (Deng & Gu, 2021) | 95.46 | 32, 64 | 84.06, 92.48 |
| | SNNC-AP(Li et al., 2021a) | 95.46 | 32, 64 | 94.78, 95.30 |
| | SNM (Wang et al., 2022) | 95.39 | 32 | 94.03 |
| | OPI(Bu et al., 2022) | 96.04 | 16, 32 | 90.43, 94.82 |
| | SlipReLU(Jiang et al., 2023) | 94.61 | 2, 4 | 93.97, 94.59 |
| | QCFS(Bu et al., 2023) | 96.04 | 2, 4 | 91.75, 93.83 |
| | SGDND(Oh & Lee, 2024) | 96.82 | 16, 32 | 80.74, 96.29 |
| | **ours** | 95.90 | **2.76** | **95.29** |
| ResNet-20 | TSC (Han & Roy, 2020) | 91.47 | 64 | 69.38 |
| | OPI(Bu et al., 2022) | 92.74 | 16, 32 | 87.22, 91.88 |
| | SlipReLU(Jiang et al., 2023) | 92.96 | 8, 16 | 86.66, 92.13 |
| | QCFS(Bu et al., 2023) | 91.77 | 4, 8 | 83.75, 89.55 |
| | **ours** | 91.52 | **3.74** | **91.13** |

## 4.2 Mixed-Timestep SNNs

**Motivation** A key insight in spiking neural networks (SNNs) is that different layers exhibit varying sensitivities to the number of timesteps. When all layers are constrained to use the same number of timesteps, model performance is often suboptimal due to this uniform limitation. As a result, attempting to enhance accuracy by uniformly increasing timesteps across layers can lead to substantial inference delays on the order of $O(NT)$, where $N$ is the number of layers and $T$ is the timestep count. To simultaneously improve performance and minimize inference latency, it is essential to allocate different timesteps to different layers.

Theorem 1 establishes the relationship between quantization bit-width in ANNs and timesteps in SNNs. Mixed-precision quantization assigns varying bit-widths across layers, denoted as $\{n_1, n_2, ..., n_l\}$, to balance performance and efficiency: higher bit-widths are allocated to more sensitive layers to mitigate performance loss, while lower bit-widths are used for less sensitive layers to reduce storage and computational overhead. Based on Theorem 1 and the conversion framework

in Section 3.3, ANNs trained with mixed-precision quantization can be efficiently converted into SNNs with mixed timesteps $\{T_1, T_2, ..., T_l\}$.

**Quantization Parameter**: In QAT for the activations of ANNs, the goal is to optimize the quantization parameter $\boldsymbol{\theta} = [d, \alpha, n]^T$, where $d \in \mathbb{R}$ represents the quantization step size, $\alpha \in \mathbb{R}$ is the maximum activation value, and $n \in \mathbb{N}$ denotes the bit-width of the quantization (Uhlich et al., 2019). Since the relationship between these three variables is given by $\alpha = d \cdot 2^n$, Only two of the three variables are needed as parameters, while the third can be derived indirectly. To overcome the challenge of non-differentiability in the rounding operation during backpropagation, we employ the Straight-Through Estimator (STE) (Bengio et al., 2013) to approximate the gradients. The parameter gradients are as follow:

**Case 1**: For $\boldsymbol{\theta} = [n, d]$, then the activation value $\alpha$ is computed as $\alpha = d \cdot 2^n$, and the quantization levels $T = 2^n$.

$$\nabla_{\boldsymbol{\theta}} Q(x;\boldsymbol{\theta}) = \begin{bmatrix} \partial_n Q(x;\boldsymbol{\theta}) \\ \partial_d Q(x;\boldsymbol{\theta}) \end{bmatrix} = \begin{cases} \begin{bmatrix} 0 \\ \frac{1}{d} \end{bmatrix} (Q(x;\boldsymbol{\theta}) - x), & 0 \le x \le \alpha \\ \begin{bmatrix} \ln 2 \cdot 2^n \cdot d \\ 2^n \end{bmatrix}, & x > \alpha \end{cases} \tag{7}$$

**Case 2**: For $\boldsymbol{\theta} = [n, \alpha]$, the step size $d$ is derived as $d = \frac{\alpha}{2^n}$, and the quantization levels $T$ remains $T = 2^n$.

$$\nabla_{\boldsymbol{\theta}} Q(x;\boldsymbol{\theta}) = \begin{bmatrix} \partial_n Q(x;\boldsymbol{\theta}) \\ \partial_\alpha Q(x;\boldsymbol{\theta}) \end{bmatrix} = \begin{cases} \begin{bmatrix} -\ln 2 \\ \frac{1}{\alpha} \end{bmatrix} (Q(x;\boldsymbol{\theta}) - x), & 0 \le x \le \alpha \\ \begin{bmatrix} 0 \\ 1 \end{bmatrix}, & x > \alpha \end{cases} \tag{8}$$

**Case 3**: For $\boldsymbol{\theta} = [d, \alpha]$, then the bit-width $n$ can be determined by $n = \log_2\left(\frac{\alpha}{d}\right)$, and the corresponding $T = \frac{\alpha}{d}$.

$$\nabla_{\boldsymbol{\theta}} Q(x;\boldsymbol{\theta}) = \begin{bmatrix} \partial_d Q(x;\boldsymbol{\theta}) \\ \partial_\alpha Q(x;\boldsymbol{\theta}) \end{bmatrix} = \begin{cases} \begin{bmatrix} \frac{1}{d} \\ 0 \end{bmatrix} (Q(x;\boldsymbol{\theta}) - x), & 0 \le x \le \alpha \\ \begin{bmatrix} 0 \\ 1 \end{bmatrix}, & x > \alpha \end{cases} \tag{9}$$

**Case 4**: When $\boldsymbol{\theta} = [t, \alpha]$, the bit-width $n$ is related to the number of time steps by $n = \log_2(t)$, and the quantization levels $T = t$.

$$\nabla_{\boldsymbol{\theta}} Q(x;\boldsymbol{\theta}) = \begin{bmatrix} \partial_t Q(x;\boldsymbol{\theta}) \\ \partial_\alpha Q(x;\boldsymbol{\theta}) \end{bmatrix} = \begin{cases} \begin{bmatrix} -\frac{1}{t} \\ \frac{1}{\alpha} \end{bmatrix} (Q(x;\boldsymbol{\theta}) - x), & 0 \le x \le \alpha \\ \begin{bmatrix} 0 \\ 1 \end{bmatrix}, & x > \alpha \end{cases} \tag{10}$$

Using the four quantization schemes outlined, we can achieve activation quantization of ANNs. In **Case 1** and **Case 2**, $n$ is chosen as the parameter, so according to Theorem 1 and the transformation framework in Section 3.2, the SNN time step $T$ corresponds to the quantization level $2^n$. In **Case 3**, the step size and threshold $[d, \alpha]$ are chosen as parameters, and the quantization bit-width needs to be indirectly derived, with the SNN time step being $\alpha/d$. For **Case 4**, the quantization level $2^n$ is treated as a whole parameter $t$, so the SNN time step $T$ can be directly obtained through training.

## 4.3 TEMPORAL ALIGNMENT

In mixed-timestep SNNs, each layer operates with different timesteps as they process information at varying temporal resolutions. Assume that the input activation of the $l+1$-th layer is $\boldsymbol{X}^{l+1}$, with the shape $\boldsymbol{X}^l \in \mathbb{R}^{T_l \times B \times C_l \times H_l \times W_l}$, where, $T_l$ is the timestep of the $l$-th layer, $B$ is the batch size, $C_l$ is the number of channels, $H_l$ and $W_l$ represent the height and width of the feature map, respectively. The expected timestep for the $l+1$-th layer is $T_{l+1}$ and in mixed-timestep SNNs, usually $T_l \neq$

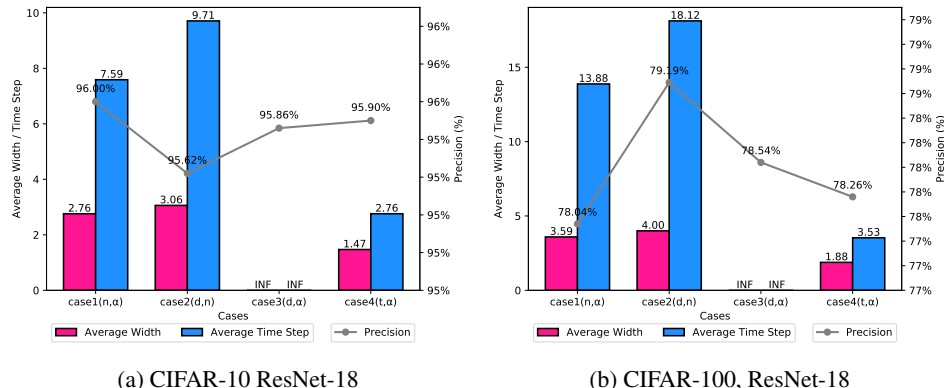

(a) CIFAR-10 ResNet-18          (b) CIFAR-100, ResNet-18

Figure 2: The average quantization bit-width, average time steps, and ANN accuracy under different quantization parameter schemes.

$T_{l+1}$, which results in the temporal dimensional mismatch. Therefore, $X_l$ must undergo a temporal alignment operation to match the timestep $T_{l+1}$. Temporal alignment can be defined as follows:

**Definition**: *Temporal alignment refers to the operation that ensures consistent time dimensions between layers with different timesteps.*

$$\hat{\boldsymbol{X}}^{l+1} = f(\boldsymbol{X}^{l+1}, T_{l+1}) \tag{11}$$

Here, $\hat{\boldsymbol{X}}^{l+1}$ is the expected input of the $l+1$-th layer and $f(\cdot)$ represents the temporal alignment function, responsible for adjusting the temporal dimension from $T_l$ to $T_{l+1}$ to ensure consistency in the temporal dimension across layers. Depending on the specific scenario, $f(\cdot)$ can involve **Temporal Expansion** (e.g., *Averaging* or *Replication*) or **Temporal Reduction** (e.g., *Averaging* or *Truncation*) to align the timesteps between successive layers. Specifically, equation 11 can be written as : $\hat{\boldsymbol{X}}^{l+1}[t] = \sum_{i=1}^{T_l} \boldsymbol{w}(t,i)\boldsymbol{X}^{l+1}[i], \quad t = 1, 2, \ldots, T_{l+1}$, where $\boldsymbol{w}(t,i)$ is the $f(\cdot)$ function that adjusts the contribution of the $i$-th timestep of $\boldsymbol{X}^l$ to the new timestep $t$ in $\hat{\boldsymbol{X}}^{l+1}$. The specific form of $\boldsymbol{w}(t,i)$ depends on the temporal alignment method: *Replication*: $\boldsymbol{w}(t,i)$ selects the nearest corresponding timestep based on integer indexing. *Averaging*: $\boldsymbol{w}(t,i) = \frac{1}{T_l}$ evenly distributes the influence of all timesteps from the previous layer. *Truncation*: $\boldsymbol{w}(t,i) = 1$ if $t \leq T_{l+1}$ and $i = t$, otherwise $\boldsymbol{w}(t,i) = 0$. Thus, truncation is effectively retaining the first $T_{l+1}$ time steps and discarding the remaining ones from $T_l$.

**Temporal Expansion**: If $T_1 < T_{l+1}$, timestep expansion must be used to increase the temporal resolution. For example , given the input activation $\boldsymbol{X}^l$, with $T_l = 4$, and the next layer $T_{l+1} = 8$, each timestep can be repeated twice by using timesteps *replication* method, which simply replicates timesteps to match the required timesteps of the subsequent layer. However, when $(T_{l+1} \mod T_l) \neq 0$, simple timestep replication leads to uneven distribution of time information. For instance, when $T_l = 4$ and $T_{l+1} = 5$, it becomes impossible to evenly distribute the repeated time steps across the new time dimension. In this case, some timesteps may be excessively duplicated, while others may not receive the necessary expansion.

**Temporal Reduction**: If $T_1 > T_{l+1}$, timestep reduction can be applied to decrease the temporal dimension. For example, if $T_l = 8$ and the next layer requires $T_{l+1} = 3$, the timesteps *truncation* method will select $T_{l+1}$ timesteps of data and discard the rest, thus matching the required dimension. However, this may lead to the loss of some information contained in the discarded time steps.

**Temporal Averaging Expansion Alignment**: To achieve temporal alignment for temporal expansion and reduction, we propose using the temporal *averaging* expansion method to adjust the input time dimensions. This method averages over the time dimension, and then replicates the result to match the desired time step length, which allows both temporal expansion and reduction to be performed using the same operation. Compared to temporal *replication* and *truncation*, the temporal *averaging* expansion method integrates information from all timesteps and achieves the best accuracy. To perform *averaging* expansion temporal alignment, we compute $\bar{\boldsymbol{X}}_l$ by averaging over the

time dimension $T_l$, obtaining a tensor of shape $\bar{\boldsymbol{X}}_l \in \mathbb{R}^{B \times C_l \times H_l \times W_l}$:

$$\bar{\boldsymbol{X}}^l = \frac{1}{T_l} \sum_{t=1}^{T_l} \boldsymbol{X}^l[t] \tag{12}$$

where, $\boldsymbol{X}^l[t]$ represents the input activation at time step $t$, and $\bar{\boldsymbol{X}}^l$ denotes the average result. Then, replicate $\bar{\boldsymbol{X}}^l$ along the time dimension to match the required timestep $T_{l+1}$, expanding $\bar{\boldsymbol{X}}^l$ to the required timesteps $T_{l+1}$, resulting in a tensor $\hat{\boldsymbol{X}}_{l+1} \in \mathbb{R}^{T_{l+1} \times B \times C_l \times H_l \times W_l}$:

$$\hat{\boldsymbol{X}}_{l+1} = \bar{\boldsymbol{X}}^l \otimes \mathbf{1}_{T_{l+1},1} \tag{13}$$

where, $\mathbf{1}_{T_{l+1},1}$ represents a tensor of length $T_{l+1}$, which replicates $\bar{\boldsymbol{X}}^l$ across $T_{l+1}$ time steps. To validate the effectiveness of the temporal *averaging* expansion method, we conducted ablation experiments on temporal alignment. The results, as shown in Table 4, indicate the following: Repl represents the use of *replication* for temporal expansion, Trunc represents the use of *truncation* for temporal reduction, and Aver represents the use of *averaging*. The results demonstrate that using the *averaging* method achieves the highest accuracy for both temporal expansion and temporal compression. Additionally, we also validated the temporal *averaging* expansion method on QCFS (Bu et al., 2023), where only a few lines of code were modified, resulting in an impressive accuracy improvement of over 10%. The experimental results can be found in the appendix Table 8 .

## 4.4 INITIAL MEMBRANE POTENTIAL AND THRESHOLD CALIBRATION

Several studies have explored the initial membrane potential and residual membrane potential, often based on assumptions such as activations following a uniform distribution (Li et al., 2021a; Bu et al., 2022; Hao et al., 2023a). For low-latency conversion, even if we set $\boldsymbol{x}^{l-1} = \Phi^{l-1}(T_l)$, Equation 4 shows that the residual term error $\epsilon$ cannot be ignored.

**Theorem 2.** *When the weight matrix $\boldsymbol{W}^l$ are fixed, the residual membrane potential $\boldsymbol{v}^l(T_l)$ is related to the initial membrane potential $\boldsymbol{v}_0^l$ and threshold $\boldsymbol{v}_{th}^l$.*

This suggests that by adjusting the initial membrane potential and threshold, the residual terms can be optimized to minimize conversion errors, the proof is provided in Appendix.

$$\boldsymbol{v}^l(T_l) = \boldsymbol{v}^l(0) + \sum_{i=1}^{T_l} \boldsymbol{I}_i - \sum_{i=0}^{T_l-1} H(\boldsymbol{v}^l(i) + \boldsymbol{I}_{i+1} - \boldsymbol{v}_{th}^l) \cdot \boldsymbol{v}_{th}^l \tag{14}$$

The weights obtained from the ANN should be frozen, and only the initial membrane potential $\boldsymbol{v}^l(0)$ and threshold $\boldsymbol{v}_{th}^l$ should be trained. The loss function is defined as:

$$\mathcal{L} = \frac{1}{2} \left( \frac{1}{T_L} \sum_{t=1}^{T_L} \boldsymbol{o}_{\text{SNN}}^L(t) - \boldsymbol{o}_{\text{ANN}}^L \right)^2 \tag{15}$$

where $\boldsymbol{o}_{\text{SNN}}^L(t)$ represent the SNN output of the final layer at time $t$, and $\boldsymbol{o}_{\text{ANN}}^L$ represent the output of the ANN. The gradient of the loss function with respect to the initial membrane potential $\boldsymbol{v}^l(0)$ is:

$$\frac{\partial \mathcal{L}}{\partial \boldsymbol{v}^l(0)} = \sum_{t=1}^{T_L} \frac{\partial \mathcal{L}}{\partial \boldsymbol{o}_{\text{SNN}}^L(t)} \cdot H'(\boldsymbol{v}^L(t) - \boldsymbol{v}_{th}^L) \cdot \prod_{k=l+1}^{L} \left( H'(\boldsymbol{v}^k(t) - \boldsymbol{v}_{th}^k) \cdot \boldsymbol{W}^k \boldsymbol{v}_{th}^{k-1} \right) \tag{16}$$

where $H'(\cdot)$ is the surrogate gradient (Wu et al., 2018b) of the Heaviside function. The gradients of the loss function with respect to the threshold is as follows, with the proof provided in the Appendix.

$$\frac{\partial \mathcal{L}}{\partial \boldsymbol{v}_{th}^l} = \sum_{t=1}^{T_L} \frac{\partial \mathcal{L}}{\partial \boldsymbol{o}_{\text{SNN}}^L(t)} \prod_{k=l}^{L-1} H'(\boldsymbol{v}^k(t) - \boldsymbol{v}_{th}^k) \cdot \boldsymbol{W}^k \boldsymbol{v}_{th}^{k-1} \cdot s^l(t-1) \tag{17}$$

## 5 EXPERIMENTS

To demonstrate the effectiveness and efficiency of the proposed algorithm, we conducted experiments on the CIFAR-10 (LeCun et al., 1998), CIFAR-100 (Krizhevsky et al., 2009), and ImageNet (Deng et al., 2009) datasets. For ease of comparison with other state-of-the-art methods, we selected basic network architectures, including ResNet-18, ResNet-20, ResNet-34, and VGG-16, more configuration information for the experiments is provided in the Appendix.

### 5.1 ACCURACY AND LATENCY OF ANNs WITH FOUR PARAMETER CONFIGURATIONS

We first evaluated the performance of activation quantized ANNs using four different parameter configurations, including the average quantization bit-width of activations and their corresponding timesteps when converted to SNNs. Figure 11 illustrates the average quantization bit-width and corresponding timesteps for all layers of ResNet-20 and VGG-16 under four different quantization parameter schemes on the CIFAR-10 and CIFAR-100 datasets. The gray curve represents the accuracy of the ANN after activation quantization for each scheme. As shown in the figure 11, the accuracy differences between the four parameter schemes are within 1%, while the number of timesteps varies significantly.

In **Case 1** and **Case 2**, the variable $n$ is used as the parameter. As shown in Figure 11, the number of timesteps in these two cases ranges from 10 to over 20, indicating relatively large timesteps. In **Case 3**: $(d, \alpha)$, the average quantization bit-width and timesteps are not shown because the bit-width is derived as $n = \log_2(\alpha/d)$ and $t = \alpha/d$, and both yielded INF values in the experimental results. From Figure 11, it is clear that **Case 4**: $(t, \alpha)$ provides the optimal quantization bit-width and timesteps, with performance nearly identical to the original ANN. Therefore, we selected **Case 4** as the quantization scheme for ANN-to-SNN conversion. Figure 1 illustrates the quantization bit-width for each layer in VGG-16 for ANNs and the corresponding timesteps for SNNs on the CIFAR-10 dataset, ResNet-18 and ResNet-20 on CIFAR-100 and ImageNet provided in the Appendix.

Table 2: Comparison between the proposed method and previous works on ImageNet dataset.

| Architecture | Methods | ANN(%) | Time Steps | SNN(%) |
|---|---|---|---|---|
| | SNNC-AP(Li et al., 2021a) | 75.36 | 32, 64 | 63.64, 70.69 |
| | SNM (Wang et al., 2022) | 73.18 | 32, 64 | 64.78, 71.50 |
| | OPI(Bu et al., 2022) | 74.85 | 32, 64 | 64.70, 72.47 |
| VGG-16 | SlipReLU(Jiang et al., 2023) | 71.99 | 32, 64 | 67.48, 71.25 |
| | QCFS(Bu et al., 2023) | 74.29 | 32, 64 | 68.47, 72.85 |
| | SGDND(Oh & Lee, 2024) | 75.35 | 32, 64 | 69.16, 75.32 |
| | **ours** | 72.12 | **3.47** | **66.38** |

### 5.2 COMPARISON WITH EXISTING CONVERSION METHODS

Table 1 presents a comparison between our method and the state-of-the-art conversion methods on the CIFAR-10 and ImageNet datasets, including TSC (Han & Roy, 2020), RTS Deng & Gu (2021), RMP (Han et al., 2020), SNM(Wang et al., 2022), SNNC-AP(Li et al., 2021a), OPI(Bu et al., 2022), SlipReLU (Jiang et al., 2023), QCFS(Bu et al., 2023) and SGDND(Oh & Lee, 2024). Since our model uses mixed-timesteps and has converged to nearly optimal timesteps, we only compare it with other works that operate under similar timestep settings. The experimental results in Table 1 are based on the **Case 4** after calibration: $(t, \alpha)$ configuration, as it offers the fewest timesteps while maintaining accuracy close to the original ANN.

**CIFAR-10**:For VGG-16, our method with 2.33 timesteps outperforms the performance of RMP(Han et al., 2020), TSC(Han & Roy, 2020), RTS(Deng & Gu, 2021), which all use 64 timesteps, as well as SNM(Wang et al., 2022) and SGDND(Oh & Lee, 2024) with 32 timesteps. Compared to QCFS(Bu et al., 2023), which achieves 93.96% accuracy with 4 timesteps, we reached 94.78% top-1 accuracy with an average of 2.33 timesteps. For ResNet-18, our method achieved 95.29% top-1 accuracy with 2.76 timesteps, whereas SNNC-AP(Li et al., 2021a) required 64 timesteps to reach 95.30%, and SGDND(Oh & Lee, 2024) needed 32 timesteps. Compared to SlipReLU (Jiang et al., 2023), which achieved 93.97% accuracy with 2 timesteps, our method outperformed SlipReLU by 1.32%. For ResNet-20, our method achieved 91.13% accuracy with 3.74 timesteps, surpassing QCFS's

(Bu et al., 2023) 83.75% accuracy with 4 timesteps. Detailed comparison data for CIFAR-100 is provided in the Appendix.

**ImageNet**:We evaluated the performance of VGG-16 on large-scale datasets, and the results demonstrate that our method significantly reduces the latency compared to previous works (Wang et al., 2022; Jiang et al., 2023; Oh & Lee, 2024; Bu et al., 2023). While prior approaches typically require over 30 timesteps to achieve around 60% accuracy, our method reaches 66.38% accuracy with an average of just 3.47 timesteps.

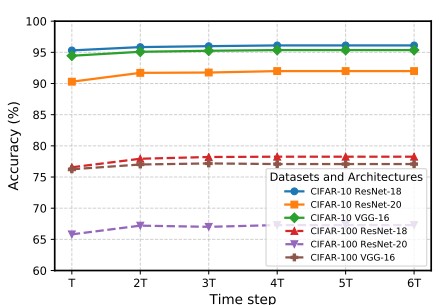

Figure 3: Temporal Scalability Analysis.

| Calibration | CIFAR-10 / CIFAR-100 / ImageNet | | |
|---|---|---|---|
| | ResNet-18 (%) | ResNet-20 (%) | VGG-16 (%) |
| QAC (Base) | 95.31 / 76.56 / - | 90.29 / 65.80 / - | 94.45 / 76.24 / 63.21 |
| QAC (Base+CAL) | **95.29 / 77.81 / -** | **91.13 / 65.39 / -** | **94.78 / 76.37 / 66.38** |

Table 3: Calibration Ablation Study.

| Temporal Alignment | CIFAR-10 / CIFAR-100 | | |
|---|---|---|---|
| | ResNet-18 (%) | ResNet-20 (%) | VGG-16 (%) |
| Repl + Aver | 95.31 / 75.50 | 89.52 / 64.86 | 94.45 / 75.70 |
| Repl + Trunc | 94.16 / 71.21 | 87.67 / 55.24 | 91.23 / 60.65 |
| Aver + Trunc | 94.16 / 75.93 | 89.20 / 62.91 | 91.23 / 74.06 |
| Aver + Aver | **95.84 / 77.93** | **91.71 / 67.20** | **95.09 / 77.01** |

Table 4: Temporal Alignment Ablation Study.

## 5.3 TEMPORAL SCALABILITY ANALYSIS

We analyzed how the accuracy of our method changes with increasing time steps to determine if performance improves or degrades over a broader temporal scale and whether the time steps from the QAC method are optimal. Experiments were conducted on ResNet-18, ResNet-20, and VGG-16 using CIFAR-10 and CIFAR-100. By doubling the time steps for each model layer, we tracked accuracy changes. As shown in Figure 3, accuracy incrementally improved with more time steps, even surpassing baseline quantized ANNs. However, beyond four times the original time steps, accuracy saturated and no longer improved.

## 5.4 CALIBRATION ABLATION STUDIES

We conducted ablation experiments to validate the effectiveness of initial membrane potential and threshold calibration. As shown in Table 3, the calibrated models (Base+CAL) achieve an approximately 1% accuracy improvement compared to pre-calibration models (Base) in CIFAR-10 and CIFAR-100, and 3% improvement on ImageNet using VGG-16 bringing their accuracy closer to that of the quantized ANN 147. The calibration module fine-tunes the initial membrane potential and threshold parameters using the output of the original ANN's final layer as labels. Notably, the calibration process does not require the training dataset and can be completed in just a few dozen epochs. Considering that the pre-calibrated SNN already delivers excellent performance under low time steps, the calibration module can be treated as an optional component in practical applications.

## 6 CONCLUSIONS

In this paper, we propose a mixed-timestep SNNs conversion method Quantization-Aware Conversion (QAC) that enables low-timestep, high-accuracy SNNs. We first demonstrate that the power of the quantization bit-width of ANN activations is equivalent to the timesteps in SNNs, showing that SNNs act as activation quantizers. Following this, we propose using mixed-precision quantization to train activation-quantized ANNs, where each layer of the network is assigned an optimal bit-width, and the converted SNNs achieve the best possible timesteps and accuracy, resulting in near-lossless conversion from ANNs. To explore the initialization of membrane potentials, we introduce a calibration method in which the final layer output of the quantized ANN serves as the target to calibrate the initial membrane potential and thresholds of the SNNs. Experimental results on CIFAR-10, CIFAR-100, and ImageNet demonstrate the effectiveness of our proposed methods.

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

# 7 APPENDIX

## 7.1 PROOF OF THEOREM

**Proof of Equation 4**: To derive the relationship between ANNs and SNNs, we begin by combining Equation 1 and 2, yielding the following expression:

$$\boldsymbol{v}^l(t) - \boldsymbol{v}^l(t-1) = \boldsymbol{W}^l \boldsymbol{v}_{th}^{l-1} \boldsymbol{s}^{l-1}(t) - \boldsymbol{v}_{th}^l s^l(t) \tag{18}$$

Next, by summing both sides of the Equation over $T_l$ timesteps, we obtain the following equation:

$$\boldsymbol{v}^l(T_l) - \boldsymbol{v}^l(0) = \sum_{t=1}^{T_l} \boldsymbol{W}^l \boldsymbol{v}_{th}^{l-1} \boldsymbol{s}^{l-1}(t) - \sum_{t=1}^{T_l} \boldsymbol{v}_{th}^l \boldsymbol{s}^l(t) \tag{19}$$

Dividing both sides by $T_l$, the Equation becomes:

$$\frac{\boldsymbol{v}^l(T_l) - \boldsymbol{v}^l(0)}{T_l} = \frac{\sum_{t=1}^{T_l} \boldsymbol{W}^l \boldsymbol{v}_{th}^{l-1} \boldsymbol{s}^{l-1}(t)}{T_l} - \frac{\sum_{t=1}^{T_l} \boldsymbol{v}_{th}^l \boldsymbol{s}^l(t)}{T_l} \tag{20}$$

Introducing $\Phi^l(T)$ to represent $\frac{\sum_{t=1}^{T_l} \boldsymbol{v}_{th}^l \boldsymbol{s}^l(t)}{T_l}$, equation 20 can be rewritten as:

$$\Phi^l(T_l) = \boldsymbol{W}^l \Phi^{l-1}(T_l) - \frac{\boldsymbol{v}^l(T_l) - \boldsymbol{v}^l(0)}{T_l} \tag{21}$$

Here, $\boldsymbol{v}^l(0)$ represents the initial membrane potential. The sum $\sum_{t=1}^{T_l} \boldsymbol{s}^{l-1}(t) = \lambda_{l-1}$, where $\lambda_l \in \{1, ..., T_l\}$. Further manipulating the equation, we obtain:

$$\boldsymbol{v}^l(T_l) - \boldsymbol{v}^l(0) = \boldsymbol{W}^l \Phi^{l-1}(T_l) \cdot T_l - \lambda_l \boldsymbol{v}_{th}^l \tag{22}$$

Solving for $\lambda_l$, we derive:

$$\lambda_l = \left\lfloor \frac{\boldsymbol{W}^l \Phi^{l-1}(T_l) \cdot T_l - \boldsymbol{v}^l(T_l) + \boldsymbol{v}^l(0)}{\boldsymbol{v}_{th}^l} \right\rceil \tag{23}$$

To ensure that $\lambda_l$ remains within a valid range, we apply the clipping operation:

$$\lambda_l = \text{clip}\left( \left\lfloor \frac{\boldsymbol{W}^l \Phi^{l-1}(T_l) \cdot T_l - \boldsymbol{v}^l(T_l) + \boldsymbol{v}^l(0)}{\boldsymbol{v}_{th}^l} \right\rceil, 0, T_l \right) \tag{24}$$

Finally, the Equation for $\Phi^l(T_l)$ becomes:

$$\Phi^l(T_l) = \frac{\boldsymbol{v}_{th}^l}{T_l} \text{clip}\left( \left\lfloor \frac{\boldsymbol{W}^l \Phi^{l-1}(T_l) \cdot T_l - \boldsymbol{v}^l(T_l) + \boldsymbol{v}^l(0)}{\boldsymbol{v}_{th}^l} \right\rceil, 0, T_l \right) \tag{25}$$

**Proof of Theorem 2**: Proof by Mathematical Induction:

We aim to prove the following recurrence relation for the membrane potential $\boldsymbol{v}^l(T_l)$ at any time step $t$:

$$\boldsymbol{v}^l(T_l) = \boldsymbol{v}^l(0) + \sum_{i=1}^{T_l} \boldsymbol{I}_i - \sum_{i=0}^{T_{l-1}} H(\boldsymbol{v}^l(i) + \boldsymbol{I}_{i+1} - \boldsymbol{v}_{th}^l)\boldsymbol{v}_{th}^l \tag{26}$$

where $\boldsymbol{I}_i = \frac{\boldsymbol{W}^l \boldsymbol{v}_{th}^{l-1} \boldsymbol{s}^{l-1}(t)}{T_l}$ represents the input current at time step $i$, $\boldsymbol{v}_{th}^l$ is the threshold potential, and $H(\cdot)$ is the Heaviside step function, which represents the occurrence of a spike (i.e., $H(x) = 1$ if $x > 0$ and $H(x) = 0$ otherwise).

Base Case: $t = 0$ At time step $t = 0$, the membrane potential is simply the initial value, as specified by the boundary condition:

$$\boldsymbol{v}^l(0) = \boldsymbol{v}^l(0) \tag{27}$$

This clearly holds true since $\boldsymbol{v}^l(0)$ is the membrane potential at $t = 0$ by definition.

Inductive Hypothesis:Assume that the formula holds for an arbitrary time step $t$, that is,

$$\boldsymbol{v}^l(T_l) = \boldsymbol{v}^l(0) + \sum_{i=1}^{t} \boldsymbol{I}_i - \sum_{i=0}^{t-1} H(\boldsymbol{v}^l(i) + \boldsymbol{I}_{i+1} - \boldsymbol{v}_{th}^l)\boldsymbol{v}_{th}^l \tag{28}$$

Inductive Step: Prove that the formula holds for $t + 1$

To prove the formula for $t + 1$, we use the recurrence relation that defines the membrane potential at time $t + 1$ as:

$$\boldsymbol{v}^l(t+1) = \boldsymbol{v}^l(T_l) + \boldsymbol{I}_{t+1} - H(\boldsymbol{v}^l(T_l) + \boldsymbol{I}_{t+1} - \boldsymbol{v}_{th}^l)\boldsymbol{v}_{th}^l \tag{29}$$

Substitute the expression for $\boldsymbol{v}^l(T_l)$ from the inductive hypothesis:

$$\boldsymbol{v}^l(t+1) = \left(\boldsymbol{v}^l(0) + \sum_{i=1}^{t} \boldsymbol{I}_i - \sum_{i=0}^{t-1} H(\boldsymbol{v}^l(i) + \boldsymbol{I}_{i+1} - \boldsymbol{v}_{th}^l)\boldsymbol{v}_{th}^l\right) + \boldsymbol{I}_{t+1} - H(\boldsymbol{v}^l(T_l) + \boldsymbol{I}_{t+1} - \boldsymbol{v}_{th}^l)\boldsymbol{v}_{th}^l \tag{30}$$

Now, simplifying this expression:

$$\boldsymbol{v}^l(t+1) = \boldsymbol{v}^l(0) + \sum_{i=1}^{t+1} \boldsymbol{I}_i - \sum_{i=0}^{t} H(\boldsymbol{v}^l(i) + \boldsymbol{I}_{i+1} - \boldsymbol{v}_{th}^l)\boldsymbol{v}_{th}^l \tag{31}$$

To reformulate the given Equation and demonstrate how the membrane potential $\boldsymbol{v}^l(T_l)$ depends on the initial potential $\boldsymbol{v}^l(0)$ when the input $\boldsymbol{I}_i$ and threshold $\boldsymbol{v}_{th}^l$, we start by expanding $\boldsymbol{v}^l(i)$ recursively based on the Equation 31. Expanding $\boldsymbol{v}^l(i)$ Recursively: for $\boldsymbol{v}^l(i)$, according to the same equation, it can be expressed as:

$$\boldsymbol{v}^l(i) = \boldsymbol{v}^l(0) + \sum_{j=1}^{i} \boldsymbol{I}_j - \sum_{j=0}^{i-1} H(V^l(j) + \boldsymbol{I}_{j+1} - \boldsymbol{v}_{th}^l)\boldsymbol{v}_{th}^l \tag{32}$$

Substituting this expression for $\boldsymbol{v}^l(i)$ back into the original Equation results in the following expanded form:

$$\boldsymbol{v}^l(T_l) = \boldsymbol{v}^l(0) + \sum_{i=1}^{t} \boldsymbol{I}_i - \sum_{i=0}^{t-1} H\left(\boldsymbol{v}^l(0) + \sum_{j=1}^{i} \boldsymbol{I}_j - \sum_{j=0}^{i-1} H(V^l(j) + \boldsymbol{I}_{j+1} - \boldsymbol{v}_{th}^l)\boldsymbol{v}_{th}^l + \boldsymbol{I}_{i+1} - \boldsymbol{v}_{th}^l\right)\boldsymbol{v}_{th}^l \tag{33}$$

Although this expression becomes complex, it highlights that $\boldsymbol{v}^l(T_l)$ can be represented in terms of the initial membrane potential $\boldsymbol{v}^l(0)$, the accumulated inputs $I$, and the reset values $\boldsymbol{v}_{th}^l$ triggered by the threshold exceeding events governed by the Heaviside function $H(\cdot)$.

## 7.2 GRADIENT CALCULATION FOR EQUATION 16 AND 17

In SNNs, the membrane potential and spike function are updated according to the following equations:

Membrane Potential Update Equation:

$$\boldsymbol{v}^l(t) = \boldsymbol{v}^l(t-1) + \boldsymbol{W}^l \boldsymbol{v}_{th}^{l-1} \boldsymbol{s}^{l-1}(t) \tag{34}$$

where $\boldsymbol{v}^l(t)$ represents the membrane potential of the neurons in layer $l$ at time $t$, $\boldsymbol{W}^l$ is the synaptic weight matrix, $\boldsymbol{v}_{th}^{l-1}$ is the threshold voltage of layer $l-1$, and $\boldsymbol{s}^{l-1}(t)$ is the spike output from layer $l-1$ at time $t$.

Spike Function:

$$\boldsymbol{s}^l(t) = H(\boldsymbol{v}^l(t) - \boldsymbol{v}_{th}^l) \tag{35}$$

where $H$ is the Heaviside step function, and $\boldsymbol{v}_{th}^l$ is the threshold voltage for layer $l$.

Soft Reset Mechanism:

$$\boldsymbol{v}^l(t) = \boldsymbol{v}^l(t) - \boldsymbol{v}_{th}^l \boldsymbol{s}^l(t) \tag{36}$$

In the context of training SNNs, the initial membrane potential $\boldsymbol{v}^m(0)$ is treated as an optimizable parameter. The backpropagation of gradients is essential for adjusting these parameters. Below, we derive the gradient of the spike function $\boldsymbol{s}^l(t)$ with respect to the initial membrane potential $\boldsymbol{v}^m(0)$ for both cases where $l = m$ and $l \neq m$.

**1. Gradient Calculation for $l = m$**

When $l = m$, the gradient is confined within the same layer, with no inter-layer propagation required.

First, the gradient of the spike function with respect to the membrane potential $\boldsymbol{v}^l(t)$ is given by:

$$\frac{\partial \boldsymbol{s}^l(t)}{\partial \boldsymbol{v}^l(t)} = H'(\boldsymbol{v}^l(t) - \boldsymbol{v}_{th}^l) \tag{37}$$

where $H'(v)$ is the surrogate gradient of the Heaviside function. Since the Heaviside function is not differentiable, a smooth approximation (e.g., tanh or piecewise linear function (Wu et al., 2018b)) is typically used to compute its derivative during backpropagation.

Next, we consider the dependency of $\boldsymbol{v}^l(t)$ on the initial membrane potential $\boldsymbol{v}^l(0)$. According to the membrane potential update equation:

$$\boldsymbol{v}^l(t) = \boldsymbol{v}^l(t-1) + \boldsymbol{W}^l \boldsymbol{v}_{th}^{l-1} \boldsymbol{s}^{l-1}(t) \tag{38}$$

we observe the recurrence relation:

$$\frac{\partial \boldsymbol{v}^l(t)}{\partial \boldsymbol{v}^l(t-1)} = 1 \tag{39}$$

Thus, the gradient of $\boldsymbol{v}^l(t)$ with respect to $\boldsymbol{v}^l(0)$ is:

$$\frac{\partial \boldsymbol{v}^l(t)}{\partial \boldsymbol{v}^l(0)} = 1 \tag{40}$$

Finally, the gradient of the spike function with respect to the initial membrane potential $\boldsymbol{v}^l(0)$ is:

$$\frac{\partial \boldsymbol{s}^l(t)}{\partial \boldsymbol{v}^l(0)} = H'(\boldsymbol{v}^l(t) - \boldsymbol{v}_{th}^l) \tag{41}$$

The direct gradient of the spike function $\boldsymbol{s}^l(t)$ with respect to the threshold $\boldsymbol{v}_{th}^l$ is given by:

$$\frac{\partial \boldsymbol{s}^l(t)}{\partial \boldsymbol{v}_{th}^l} = -H'(\boldsymbol{v}^l(t) - \boldsymbol{v}_{th}^l) \tag{42}$$

Since the membrane potential $\boldsymbol{v}^l(t-1)$ depends on the spike function $s^l(t-1)$ at the previous time step, we need to compute the gradient with respect to $\boldsymbol{v}_{th}^l$ through the membrane potential at time $t-1$:

$$\frac{\partial \boldsymbol{v}^l(t-1)}{\partial \boldsymbol{v}_{th}^l} = -s^l(t-1) \tag{43}$$

Therefore, combining the direct and indirect gradients, the total gradient of $\boldsymbol{s}^l(t)$ with respect to $\boldsymbol{v}_{th}^l$ when $l = m$ is:

$$\frac{\partial \boldsymbol{s}^l(t)}{\partial \boldsymbol{v}_{th}^l} = -H'(\boldsymbol{v}^l(t) - \boldsymbol{v}_{th}^l)(1 + s^l(t-1)) \tag{44}$$

**2. Gradient Calculation for $l \neq m$**

When $l \neq m$, the gradient must propagate through multiple layers from layer $l$ back to layer $m$. In this case, we apply the chain rule for backpropagation across layers.

The gradient of the spike function $\boldsymbol{s}^l(t)$ in layer $l$ with respect to the spike function $\boldsymbol{s}^{l-1}(t)$ in the previous layer is given by:

$$\frac{\partial \boldsymbol{s}^l(t)}{\partial \boldsymbol{s}^{l-1}(t)} = \frac{\partial \boldsymbol{s}^l(t)}{\partial \boldsymbol{v}^l(t)} \cdot \frac{\partial \boldsymbol{v}^l(t)}{\partial \boldsymbol{s}^{l-1}(t)} \tag{45}$$

Substituting the known terms:

$$\frac{\partial \boldsymbol{s}^l(t)}{\partial \boldsymbol{v}^l(t)} = H'(\boldsymbol{v}^l(t) - \boldsymbol{v}_{th}^l) \tag{46}$$

and

$$\frac{\partial \boldsymbol{v}^l(t)}{\partial \boldsymbol{s}^{l-1}(t)} = \boldsymbol{W}^l \boldsymbol{v}_{th}^{l-1} \tag{47}$$

we obtain:

$$\frac{\partial \boldsymbol{s}^l(t)}{\partial \boldsymbol{s}^{l-1}(t)} = H'(\boldsymbol{v}^l(t) - \boldsymbol{v}_{th}^l) \cdot \boldsymbol{W}^l \boldsymbol{v}_{th}^{l-1} \tag{48}$$

Multi-Layer Gradient Propagation:

By recursively applying the chain rule across layers, we derive the following:

$$\frac{\partial \boldsymbol{s}^l(t)}{\partial \boldsymbol{v}^m(0)} = \prod_{k=m+1}^{l} \left( \frac{\partial \boldsymbol{s}^k(t)}{\partial \boldsymbol{s}^{k-1}(t)} \right) \cdot \frac{\partial \boldsymbol{s}^m(t)}{\partial \boldsymbol{v}^m(0)} \tag{49}$$

For the same layer $m$, the gradient of the spike function with respect to the initial membrane potential is:

$$\frac{\partial \boldsymbol{s}^m(t)}{\partial \boldsymbol{v}^m(0)} = H'(\boldsymbol{v}^m(t) - \boldsymbol{v}_{th}^m) \tag{50}$$

Thus, the complete recursive gradient for layers $l > m$ is:

$$\frac{\partial \boldsymbol{s}^l(t)}{\partial \boldsymbol{v}^m(0)} = \prod_{k=m+1}^{l} \left( H'(\boldsymbol{v}^k(t) - \boldsymbol{v}_{th}^k) \cdot \boldsymbol{W}^k \boldsymbol{v}_{th}^{k-1} \right) \cdot H'(\boldsymbol{v}^m(t) - \boldsymbol{v}_{th}^m) \tag{51}$$

By recursively expanding this gradient expression down to layer $m$, we obtain the following gradient:

$$\frac{\partial \boldsymbol{s}^l(t)}{\partial \boldsymbol{v}_{th}^m} = \prod_{k=m}^{l-1} H'(\boldsymbol{v}^k(t) - \boldsymbol{v}_{th}^k) \boldsymbol{W}^k \boldsymbol{v}_{th}^{k-1} \cdot (-\boldsymbol{s}^m(t-1)) \tag{52}$$

### 3. Gradient of the Loss with Respect to Initial Membrane Potential and threshold

The loss function of the SNN is based on the mean squared error (MSE) between the average output of the final layer over $T$ time steps and the corresponding output of ANN. Let $o_{\text{SNN}}^L(t)$ represent the SNN output of the final layer at time $t$, and $o_{\text{ANN}}^L$ represent the output of the ANN. The loss function is defined as:

$$\mathcal{L} = \frac{1}{2} \left( \frac{1}{T} \sum_{t=1}^{T} o_{\text{SNN}}^L(t) - o_{\text{ANN}}^L \right)^2 \tag{53}$$

The gradient of the loss function $\mathcal{L}$ with respect to the SNN output $o_{\text{SNN}}^L(t)$ is:

$$\frac{\partial \mathcal{L}}{\partial o_{\text{SNN}}^L(t)} = \frac{1}{T} \left( \frac{1}{T} \sum_{t=1}^{T} o_{\text{SNN}}^L(t) - o_{\text{ANN}}^L \right) \tag{54}$$

Since the SNN output $o_{\text{SNN}}^L(t)$ depends on the membrane potential $\boldsymbol{v}^L(t)$, the gradient of the loss function with respect to $\boldsymbol{v}^L(t)$ is:

$$\frac{\partial \mathcal{L}}{\partial \boldsymbol{v}^L(t)} = \frac{\partial \mathcal{L}}{\partial o_{\text{SNN}}^L(t)} \cdot \frac{\partial o_{\text{SNN}}^L(t)}{\partial \boldsymbol{v}^L(t)} \tag{55}$$

Here, $\frac{\partial o_{\text{SNN}}^L(t)}{\partial \boldsymbol{v}^L(t)} = H'(\boldsymbol{v}^L(t) - \boldsymbol{v}_{th}^L)$, where $H'(v)$ is the surrogate gradient of the Heaviside step function. Therefore, the gradient becomes:

$$\frac{\partial \mathcal{L}}{\partial \boldsymbol{v}^L(t)} = \frac{1}{T} \left( \frac{1}{T} \sum_{t=1}^{T} o_{\text{SNN}}^L(t) - o_{\text{ANN}}^L \right) \cdot H'(\boldsymbol{v}^L(t) - \boldsymbol{v}_{th}^L) \tag{56}$$

To compute the gradient with respect to $\boldsymbol{v}^l(0)$ for each layer $l \neq L$, we propagate the gradient backwards from the final layer. Using the chain rule, we obtain:

$$\frac{\partial \mathcal{L}}{\partial \boldsymbol{v}^l(0)} = \sum_{t=1}^{T} \frac{\partial \mathcal{L}}{\partial \boldsymbol{v}^L(t)} \cdot \frac{\partial \boldsymbol{v}^L(t)}{\partial s^{L-1}(t)} \cdot \frac{\partial s^{L-1}(t)}{\partial \boldsymbol{v}^l(0)} \tag{57}$$

Where $\frac{\partial \boldsymbol{v}^L(t)}{\partial s^{L-1}(t)} = \boldsymbol{W}^L V_{\text{th}}^{L-1}$, and $\frac{\partial s^{L-1}(t)}{\partial \boldsymbol{v}^l(0)}$ can be computed using the chain rule for cross-layer backpropagation.

The cross-layer gradient is propagated recursively as:

$$\frac{\partial s^l(t)}{\partial \boldsymbol{v}^m(0)} = \prod_{k=m+1}^{l} \left( H'(\boldsymbol{v}^k(t) - \boldsymbol{v}_{th}^k) \cdot \boldsymbol{W}^k \boldsymbol{v}_{th}^{k-1} \right) \cdot H'(\boldsymbol{v}^m(t) - \boldsymbol{v}_{th}^m) \tag{58}$$

Therefore, the gradient of the loss function with respect to the initial membrane potential $\boldsymbol{v}^l(0)$ is:

$$\frac{\partial \mathcal{L}}{\partial \boldsymbol{v}^l(0)} = \sum_{t=1}^{T} \frac{1}{T} \left( \frac{1}{T} \sum_{t=1}^{T} o_{\text{SNN}}^L(t) - o_{\text{ANN}}^L \right) \cdot H'(\boldsymbol{v}^L(t) - \boldsymbol{v}_{th}^L) \cdot \prod_{k=l+1}^{L} \left( H'(\boldsymbol{v}^k(t) - \boldsymbol{v}_{th}^k) \cdot \boldsymbol{W}^k \boldsymbol{v}_{th}^{k-1} \right)$$
$$\tag{59}$$

The gradient of the loss function with respect to $\boldsymbol{v}_{th}^l$ is:

$$\frac{\partial \mathcal{L}}{\partial \boldsymbol{v}_{th}^l} = \sum_{t=1}^{T} \frac{\partial \mathcal{L}}{\partial y^{\text{SNN}}(t)} \cdot \frac{\partial y^{\text{SNN}}(t)}{\partial s^L(t)} \cdot \prod_{k=l}^{L} \frac{\partial \boldsymbol{s}^k(t)}{\partial \boldsymbol{v}_{th}^l} \tag{60}$$

From the previous derivation, the recursive relation for the gradient of the spike function with respect to the threshold $\boldsymbol{v}_{th}^l$ is:

$$\frac{\partial s^L(t)}{\partial \boldsymbol{v}_{th}^l} = \prod_{k=l}^{L-1} H'(\boldsymbol{v}^k(t) - \boldsymbol{v}_{th}^k)\boldsymbol{W}^k \boldsymbol{v}_{th}^{k-1} \cdot (-s^l(t-1)) \tag{61}$$

Therefore, the complete gradient is:

$$\frac{\partial \mathcal{L}}{\partial \boldsymbol{v}_{th}^l} = -\sum_{t=1}^{T} \frac{1}{T} \left( \frac{1}{T} \sum_{t=1}^{T} y^{\text{SNN}}(t) - y^{\text{ANN}} \right) \prod_{k=l}^{L-1} H'(\boldsymbol{v}^k(t) - \boldsymbol{v}_{th}^k)\boldsymbol{W}^k \boldsymbol{v}_{th}^{k-1} \cdot (-s^l(t-1)) \tag{62}$$

---

**Algorithm 1** Algorithm for ANN-to-SNN conversion.

---

**Require:** ANN model $M_{ANN}(\boldsymbol{x}; \boldsymbol{W})$ without pretrained weight $\boldsymbol{W}$; Dataset $D$; Quantization parameters $\boldsymbol{\theta} = [\alpha, d, n]^T$;
**Ensure:** $M_{SNN}(\boldsymbol{x}; \tilde{\boldsymbol{W}})$
1: **for** $l = 1$ to $M_{ANN}.layers$ **do**
2:     **if** ReLU activation **then**
3:         Replace ReLU($\boldsymbol{x}$) by QAC($\boldsymbol{x}; \boldsymbol{\theta}$)
4:     **end if**
5:     **if** MaxPooling layer **then**
6:         Replace MaxPooling layer by AvgPooling layer
7:     **end if**
8: **end for**
9: **for** $e = 1$ to epochs **do**
10:     **for** length of Dataset $D$ **do**
11:         Sample minibatch $(\boldsymbol{x}^0, \boldsymbol{y})$ from $D$
12:         **for** $l = 1$ to $M_{ANN}.layers$ **do**
13:             $\boldsymbol{x}^l = QAC(\boldsymbol{W}^l \boldsymbol{x}^{l-1}; \boldsymbol{\theta})$
14:         **end for**
15:         Loss = CrossEntropy($\boldsymbol{x}^l, \boldsymbol{y}$)
16:         **for** $l = 1$ to $M_{ANN}.layers$ **do**
17:             $\boldsymbol{W}^l \leftarrow \boldsymbol{W}^l - \epsilon \frac{\partial Loss}{\partial \boldsymbol{W}^l}$
18:             $\boldsymbol{\theta}^l \leftarrow \boldsymbol{\theta}^l - \epsilon \frac{\partial Loss}{\partial \boldsymbol{\theta}^l}$
19:         **end for**
20:     **end for**
21: **end for**
22: **for** $l = 1$ to $M_{ANN}.layers$ **do**
23:     $M_{SNN}.\tilde{\boldsymbol{W}}^l \leftarrow M_{ANN}.\boldsymbol{W}^l$
24:     $M_{SNN}.\boldsymbol{\theta}^l \leftarrow M_{ANN}.\boldsymbol{\theta}^l$
25: **end for**
26: **for** $e = 1$ to epochs **do**
27:     Loss = MSE($\boldsymbol{y}_{SNN}, \boldsymbol{y}_{ANN}$)
28:     **for** $l = 1$ to $M_{SNN}.layers$ **do**
29:         $\boldsymbol{v}^l(0) \leftarrow \boldsymbol{v}^l(0) - \epsilon \frac{\partial Loss}{\partial \boldsymbol{v}^l(0)}$
30:         $\boldsymbol{v}_{th}^l \leftarrow \boldsymbol{v}_{th}^l - \epsilon \frac{\partial Loss}{\partial \boldsymbol{v}_{th}^l}$
31:     **end for**
32: **end for**
33: **return** $M_{SNN}$

---

## 7.3 TEMPORAL SCALABILITY ANALYSIS.

To verify whether the QAC method can achieve better performance with more time steps, we conducted temporal expansion experiments. The results, shown in Table 5, detail the base time steps T for various models on different datasets. For example, on CIFAR-100, the base time steps T for ResNet-18, ResNet-20, and VGG-16 are 3.52, 4.58, and 3.67, respectively, which are consistent with the data in Table 7. Similar experiments for time step expansion were conducted on CIFAR-10. The results indicate that our method converges to near-optimal accuracy within a short time. As the time steps double, the accuracy increases slightly. However, when the time steps are increased to 3–4 times the original, the accuracy saturates and no longer changes significantly.

Table 5: Temporal Scalability Analysis.

| Datasets | Architecture | T | 2T | 3T | 4T | 5T | 6T |
|----------|--------------|------|------|------|------|------|------|
| CIFAR-10 | ResNet-18 | 95.31% | 95.84% | 95.99% | 96.11% | 96.11% | 96.11% |
| | ResNet-20 | 90.29% | 91.71% | 91.76% | 91.99% | 91.99% | 91.99% |
| | VGG-16 | 94.45% | 95.09% | 95.25% | 95.36% | 95.36% | 95.36% |
| CIFAR-100 | ResNet-18 | 76.56% | 77.93% | 78.21% | 78.26% | 78.26% | 78.26% |
| | ResNet-20 | 65.80 % | 67.20% | 67.01% | 67.29% | 67.29% | 67.29% |
| | VGG-16 | 76.24% | 77.01% | 77.18% | 77.08% | 77.08% | 77.08% |

## 7.4 COMPARISON WITH OTHER DIRECTLY TRAINING METHODS.

Table 6 compares the performance of QAC (our method) with other directly trained methods using surrogate gradients across three datasets: CIFAR-10, CIFAR-100, and ImageNet-1k. QAC demonstrates competitive accuracy with significantly fewer time steps compared to other methods. For CIFAR-10, QAC achieves 91.13% accuracy on ResNet-20 with just 3.74 time steps, and 94.78% accuracy on VGG-16 with 2.33 time steps, outperforming most surrogate gradient methods in efficiency. On CIFAR-100, QAC achieves 77.81% on ResNet-18 with 3.52 time steps and 76.37% on VGG-16 with 3.67 time steps, showing strong performance relative to surrogate gradient methods. For ImageNet-1k, QAC achieves 66.38% on VGG-16 with 3.47 time steps, demonstrating a balance of accuracy and efficiency. Overall, QAC achieves near-optimal accuracy with fewer time steps, highlighting its computational efficiency and scalability across datasets and architectures.

## 7.5 EXPERIMENT RESULTS ON CIFAR-100 DATASET

Table 7 shows that our method achieves competitive or superior SNN accuracy with significantly fewer time steps across VGG-16, ResNet-18, and ResNet-20 on CIFAR-100. For VGG-16, it achieves the highest accuracy 76.37% with just 3.67 time steps, outperforming methods like RTS and SNNC-AP, which require up to 256 and 32 steps. Similarly, for ResNet-18 and ResNet-20, it achieves strong accuracy 75.51% and 65.39% with only 3.52 and 4.58 steps. These results highlight our method's efficiency and rapid convergence to near-optimal accuracy.

## 7.6 TEMPORAL AVERAGING EXPANSION ALIGNMENT

Table 8 demonstrates that QCFS with temporal averaging (QCFS+aver) consistently outperforms standard QCFS across all architectures, datasets, and quantization levels, particularly in low time-step settings and at higher quantization levels ($L = 4, 8, 16$). Temporal averaging significantly enhances accuracy, especially when time steps $T$ are limited, achieving comparable or higher performance with fewer steps. For example, in ResNet-20 on CIFAR-10, QCFS+aver maintains over 90% accuracy across higher $T$ values even at $L = 16$, while QCFS shows substantial accuracy drops. Similarly, in VGG-16 on CIFAR-100, QCFS+aver shows strong improvements under challenging settings, particularly at high quantization levels. These results highlight the effectiveness of temporal averaging in boosting performance and computational efficiency.

Table 6: Comparison with other directly training methods.

| Dataset | Method | Type | Architecture | Time-steps | Accuracy (%) |
|---|---|---|---|---|---|
| CIFAR-10 | tdBN (Zheng et al., 2021) | Surrogate Gradient | ResNet-18 | 4 | 92.92 |
| | Dspike (Li et al., 2021b) | Surrogate Gradient | ResNet-18 | 4 | 93.66 |
| | TET (Deng et al., 2022) | Surrogate Gradient | ResNet-19 | 4 | 94.44 |
| | GLIF (Yao et al., 2022) | Surrogate Gradient | ResNet-19 | 2, 4, 6 | 94.15, 94.67, 94.88 |
| | RMP-Loss (Guo et al., 2023) | Surrogate Gradient | ResNet-20 | 4 | 91.89 |
| | PSN (Fang et al., 2024) | Surrogate Gradient | Modified PLIF Net | 4 | 95.32 |
| | TAB (Jiang et al., 2024a) | Surrogate Gradient | VGG-9 | 4 | 93.41 |
| | | | ResNet-19 | 2, 4, 6 | 94.73, 94.76, 94.81 |
| | **QAC(Ours)** | ANN-to-SNN | ResNet-18 | 2.76 | 95.29 |
| | | | ResNet-20 | 3.74 | 91.13 |
| | | | VGG-16 | 2.33 | 94.78 |
| CIFAR-100 | Dspike (Li et al., 2021b) | Surrogate Gradient | ResNet-18 | 4 | 73.35 |
| | TET (Deng et al., 2022) | Surrogate Gradient | ResNet-19 | 4 | 74.47 |
| | GLIF (Yao et al., 2022) | Surrogate Gradient | ResNet-19 | 2, 4, 6 | 75.48, 77.05, 77.35 |
| | RMP-Loss (Guo et al., 2023) | Surrogate Gradient | ResNet-19 | 2, 4, 6 | 74.66, 78.28, 78.98 |
| | TAB (Jiang et al., 2024a) | Surrogate Gradient | VGG-9 | 4 | 75.89 |
| | | | ResNet-19 | 2, 4, 6 | 76.31, 76.81, 76.82 |
| | **QAC(Ours)** | ANN-to-SNN | ResNet-18 | 3.52 | 77.81 |
| | | | ResNet-20 | 4.58 | 65.39 |
| | | | VGG-16 | 3.67 | 76.37 |
| ImageNet-1k | tdBN (Zheng et al., 2021) | Surrogate Gradient | ResNet-34 | 6 | 63.72 |
| | SEW ResNet (Fang et al., 2021) | Surrogate Gradient | SEW ResNet-34 | 6 | 67.04 |
| | TET (Deng et al., 2022) | Surrogate Gradient | SEW ResNet-34 | 4 | 68.00 |
| | GLIF (Yao et al., 2022) | Surrogate Gradient | ResNet-34 | 6 | 67.52 |
| | RMP-Loss (Guo et al., 2023) | Surrogate Gradient | ResNet-34 | 4 | 65.17 |
| | TAB (Jiang et al., 2024a) | Surrogate Gradient | ResNet-34 | 2,4 | 65.94, 67.78 |
| | PSN (Fang et al., 2024) | Surrogate Gradient | SEW Resnet-34 | 4 | 70.54 |
| | PSN (Fang et al., 2024) | Surrogate Gradient | SEW Resnet-34 | 4 | 70.54 |
| | **QAC(Ours)** | ANN-to-SNN | VGG-16 | 3.47 | 66.38 |

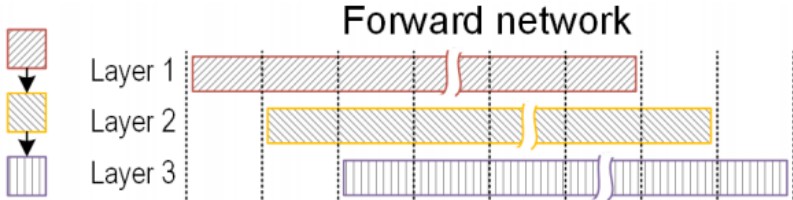

Figure 4: Neuromorphic Hardware Pipeline.

## 7.7 HARDWARE EFFICIENCY ANALYSIS

Using QAC to build mixed timestep SNNs allows them to run on SNN hardware while preserving the asynchronous nature of SNNs. SNN hardware has two mainstream implementation approaches: ANN accelerator variants and non-Von Neumann distributed multi-core architectures (e.g., TrueNorth Akopyan et al. (2015), Loihi Davies et al. (2018)) Li et al. (2024a).

ANN accelerator variants primarily achieve asynchronous computation by sending non-zero inputs to processing element PE arrays and performing spike-based matrix calculations. These accelerators only compute one part of the neural network at a time, iterating to cover the entire network. Algorithm 2 Li et al. (2024b) shows the data flow, where the timestep for each layer is $T$. For mixed-timestep SNNs, we modify the timestep $T_l$ for each layer and average the outputs of each layer along the time dimension. These two operations do not change the original data flow, allowing the model to run on this type of hardware.

In contrast, multi-core neuromorphic hardware deploys the neurons of all layers across different cores. When neurons receive spike events, they immediately perform spike-based computations, achieving asynchronous execution. The network runs on hardware in a pipelined manner. As shown

Table 7: Comparison between our method and previous works on CIFAR-100 dataset.

| Architecture | Methods | ANN | Time Steps | SNN |
|---|---|---|---|---|
| VGG-16 | RMP (Han et al., 2020) | 71.22 | 128 | 63.76 |
| | TSC (Han & Roy, 2020) | 71.22 | 128 | 69.86 |
| | RTS (Deng & Gu, 2021) | 77.89 | 256 | 73.54 |
| | SNNC-AP(Li et al., 2021a) | 77.89 | 32 | 73.55 |
| | SNM (Wang et al., 2022) | 74.13 | 32 | 71.8 |
| | OPI(Bu et al., 2022) | 76.31 | 16, 32 | 70.72, 74.82 |
| | SlipReLU(Jiang et al., 2023) | 68.46 | 4, 8 | 67.97, 69.31 |
| | QCFS(Bu et al., 2023) | 76.28 | 4, 8 | 69.62, 73.96 |
| | SGDND(Oh & Lee, 2024) | 78.28 | 16, 32 | 39.42, 76.33 |
| | **ours** | 76.53 | **3.67** | **76.37** |
| ResNet-18 | RTS* (Deng & Gu, 2021) | 77.16 | 64 | 70.12 |
| | SNNC-AP*(Li et al., 2021a) | 77.16 | 32, 64 | 76.32, 77.29 |
| | SlipReLU(Jiang et al., 2023) | 74.01 | 4, 8 | 74.89, 75.40 |
| | QCFS(Bu et al., 2023) | 78.80 | 2, 4 | 70.79, 75.67 |
| | **ours** | 78.26 | **3.52** | **77.81** |
| ResNet-20 | RMP (Han et al., 2020) | 68.72 | 128 | 57.69 |
| | TSC (Han & Roy, 2020) | 68.72 | 128 | 58.42 |
| | RTS* (Deng & Gu, 2021) | 77.16 | 64, 128 | 70.12, 75.81 |
| | SNNC-AP* (Li et al., 2021a) | 77.16 | 32, 64 | 76.32, 77.29 |
| | SNM* (Wang et al., 2022) | 78.26 | 32, 64 | 74.48, 77.59 |
| | OPI(Bu et al., 2022) | 70.43 | 32 | 67.18 |
| | SlipReLU(Jiang et al., 2023) | 68.40 | 8, 16 | 57.20, 66.61 |
| | QCFS(Bu et al., 2023) | 69.94 | 4, 8 | 34.14, 55.37 |
| | SGDND(Oh & Lee, 2024) | 81.19 | 16, 32 | 36.78, 79.13 |
| | **ours** | 66.32 | **4.58** | **65.39** |

* is not standard ResNet-18.

in Figure 4 Zhong et al. (2024), at time $T_1$, Layer 1 processes the data from Sample 1. At $T_2$, Layer 2 processes the data from Sample 1 (i.e., the output of Layer 1 at $T_1$), while Layer 1 processes the data from Sample 2.

For mixed timestep SNNs, a time alignment strategy must be used to handle the different timesteps of each layer. During pipeline execution, Layer 2 must wait until Layer 1 completes $T_{l1}$ timesteps before it can start computation. Although mixed timestep SNNs can run on this type of hardware, pipeline stalling may occur, introducing computational delays and preventing the hardware from achieving optimal performance.

## 7.8 ENERGY COMPARISON.

To compare the energy consumption of SNNs (Spiking Neural Networks) and quantized ANNs (Artificial Neural Networks), we conducted experiments on VGG-16 using the CIFAR-100 dataset. We assume all weights are represented in 8-bit integers (INT8). Based on the energy consumption of 32-bit and 8-bit fixed-point and floating-point operations provided in (Horowitz, 2014), we calculate the energy for a multiply-accumulate (MAC) operation, which is the sum of the energy required for multiplication and addition.

However, since the activations in the quantized ANN use mixed-precision quantization (with varying bit widths across layers), calculating the energy for MAC operations involves handling fixed-point multiplications with varying bit widths. To address this, we refer to the findings in (Potipireddi & Asati, 2013), which indicate that the energy consumption of an adder scales linearly with bit width, while the energy consumption of a multiplier scales quadratically with bit width . This enables us to calculate the energy consumption for MAC operations involving fixed-point numbers of varying bit widths.In contrast, SNNs only require addition operations, eliminating the challenges posed by varying bit widths in quantized ANNs.

Table 8: Comparison of QCFS Results with and without Temporal Averaging Expansion Alignment.

| | Method | T=1 | T=2 | T=3 | T=4 | T=8 | T=16 | T=32 | T=64 |
|---|---|---|---|---|---|---|---|---|---|
| | | | ResNet-20 on CIFAR-10 | | | | | | |
| L=2 | QCFS | 78.85% | 83.94% | 86.43% | 87.9% | 89.69% | 90.06% | 89.97% | 89.8% |
| | QCFS+(aver) | 78.85% | 88.58% | 89.25% | 89.31% | 89.57% | 88.96% | 88.27% | 89.77% |
| L=4 | QCFS | 62.32% | 71.67% | 78.21% | 82.5% | 89.48% | 91.83% | 92.5% | 92.59% |
| | QCFS+(aver) | 62.32% | 87.30% | 90.78% | 91.90% | 92.39% | 92.54% | 92.62% | 92.61% |
| L=8 | QCFS | 52.68% | 65.58% | 73.48% | 78.64% | 88.31% | 92.32% | 93.21% | 93.50% |
| | QCFS+(aver) | 52.69% | 83.47% | 89.67% | 91.54% | 93.21% | 93.54% | 93.64% | 93.70% |
| L=16 | QCFS | 36.45% | 47.72% | 56.95% | 65.9% | 84.12% | 91.71% | 93.22% | 93.48% |
| | QCFS+(aver) | 36.45% | 75.29% | 87.59% | 91.13% | 93.05% | 93.59% | 93.58% | 93.57% |
| | | | VGG-16 on CIFAR-10 | | | | | | |
| L=2 | QCFS | 61.45% | 71.38% | 74.57% | 75.92% | 77.38% | 77.79% | 77.79% | 77.87% |
| | QCFS+(aver) | 61.45% | 77.61% | 77.5% | 77.91% | 77.89% | 77.98% | 77.84% | 77.88% |
| L=4 | QCFS | 10.89% | 77.20% | 84.35% | 88.49% | 93.33% | 95.08% | 95.76% | 95.90% |
| | QCFS+(aver) | 10.89% | 91.02% | 94.44% | 95.33% | 95.75% | 95.94% | 96.01% | 95.99% |
| L=8 | QCFS | 72.03% | 86.62% | 92.03% | 93.46% | 95.07% | 95.70% | 95.74% | 95.77% |
| | QCFS+(aver) | 72.03% | 93.32% | 95.22% | 95.64% | 95.77% | 95.83% | 95.83% | 95.84% |
| L=16 | QCFS | 29.02% | 86.02% | 89.38% | 91.91% | 94.65% | 95.77% | 96.03% | 96.07% |
| | QCFS+(aver) | 29.02% | 92.84% | 94.66% | 95.39% | 95.85% | 96.04% | 96.03% | 96.04% |
| | | | ResNet-20 on CIFAR-100 | | | | | | |
| L=2 | QCFS | 43.71% | 44.68% | 55.64% | 58.17% | 61.18% | 62.09% | 61.93% | 61.56% |
| | QCFS+(aver) | 43.71% | 58.97% | 60.34% | 61.17% | 60.92% | 61.32% | 61.14% | 61.14% |
| L=4 | QCFS | 25.64% | 36.00% | 44.10% | 50.36% | 62.02% | 66.33% | 67.26% | 67.05% |
| | QCFS+(aver) | 25.64% | 56.56% | 63.66% | 64.67% | 66.11% | 66.55% | 66.76% | 66.50% |
| L=8 | QCFS | 11.48% | 17.11% | 23.46% | 29.94% | 51.29% | 64.65% | 68.03% | 68.62% |
| | QCFS+(aver) | 11.48% | 43.82% | 59.06% | 64.47% | 67.96% | 68.06% | 68.52% | 68.62% |
| L=16 | QCFS | 7.26% | 11.15% | 15.47% | 20.54% | 41.95% | 61.81% | 68.07% | 69.02% |
| | QCFS+(aver) | 7.26% | 32.49% | 53.15% | 61.61% | 68.28% | 69.13% | 69.23% | 69.32% |
| | | | VGG-16 on CIFAR-100 | | | | | | |
| L=2 | QCFS | 65.06% | 68.97% | 71.13% | 72.3% | 74.34% | 75.13% | 75.43% | 75.60% |
| | QCFS+(aver) | 65.06% | 73.85% | 74.34% | 74.96% | 75.56% | 75.39% | 75.54% | 75.54% |
| L=4 | QCFS | 57.57% | 64.33% | 67.93% | 70.13% | 74.75% | 76.33% | 77.01% | 77.15% |
| | QCFS+(aver) | 57.57% | 73.01% | 75.53% | 76.30% | 76.90% | 77.03% | 77.08% | 77.24% |
| L=8 | QCFS | 45.47% | 55.55% | 60.53% | 64.93% | 72.42% | 76.02% | 77.22% | 77.44% |
| | QCFS+(aver) | 45.47% | 69.88% | 74.58% | 75.7% | 75.58% | 77.15% | 77.14% | 77.11% |
| L=16 | QCFS | 28.98% | 41.11% | 48.66% | 54.41% | 67.02% | 74.39% | 76.87% | 77.56% |
| | QCFS+(aver) | 28.98% | 66.23% | 73.58% | 75.48% | 76.86% | 77.71% | 77.68% | 77.69% |

L is the quantization step in QCFS.

In contrast, SNNs only require addition operations, eliminating the challenges posed by varying bit widths in quantized ANNs.

In contrast, SNNs only require addition operations, eliminating the challenges posed by varying bit widths in quantized ANNs. With $N$ input channels, $M$ output channels, input map size $I \times I$, weight kernel size $k \times k$, and output size $O \times O$, the total FLOPS for ANN/SNN are:

$$FLOPS_{ANN} = O^2 \times N \times k^2 \times M \tag{63}$$

$$FLOPS_{SNN} = O^2 \times N \times k^2 \times M \times S_A \tag{64}$$

Where $S_A$ is the net spiking activity, representing the total number of firing neurons per layer.

For energy calculation, each MAC (for ANN) or AC (Addition operation, for SNN) is considered, as specified below:

$$E_{ANN} = \left( \sum_{i=1}^{N} FLOPS_{ANN} \right) \times E_{MAC} \tag{65}$$

**Algorithm 2** Neuron and Temporal Loops

```
1:  for o ← 0 to C_o/M do                                      ▷ Neuron Loops
2:      for h ← 0 to H_o do
3:          for w ← 0 to W_o/N do
4:              for t ← 0 to T/S do                            ▷ Temporal Loop
5:                  for k_h ← 0 to K_h do                      ▷ Spatial Loops
6:                      for k_w ← 0 to K_w do
7:                          for i ← 0 to C_i/V do
8:                              P_sum += W × I_spikes          ▷ Unrolled computation
9:                          end for
10:                     end for
11:                 end for
12:                 V_Next, O_spikes ← Node(P_sum, V_Pre)
13:             end for
14:         end for
15:     end for
16: end for
17: return V_Next, O_spikes
```

$$E_{SNN} = \left( \sum_{i=1}^{N} FLOPS_{SNN} \right) \times E_{AC} \times T_l \tag{66}$$

Where: $E_{MAC}$: Energy per MAC operation. $E_{AC}$: Energy per Addition operation. $T_l$: Number of time steps in SNN l-th layer.

## 7.9 TIME STEP VS. BIT WIDTH

The following image shows the quantization bit-width and timesteps corresponding to different training parameters used during quantization-aware training (QAT) of ResNet-18, ResNet-20, and VGG-16 on CIFAR-10, CIFAR-100, and ImageNet.

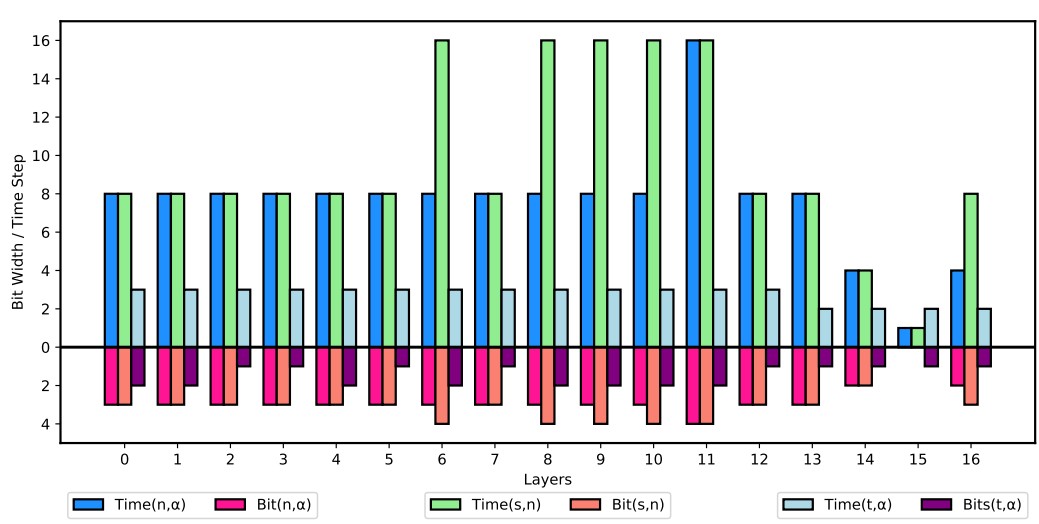

Figure 5: CIFAR-10, ResNet-18.

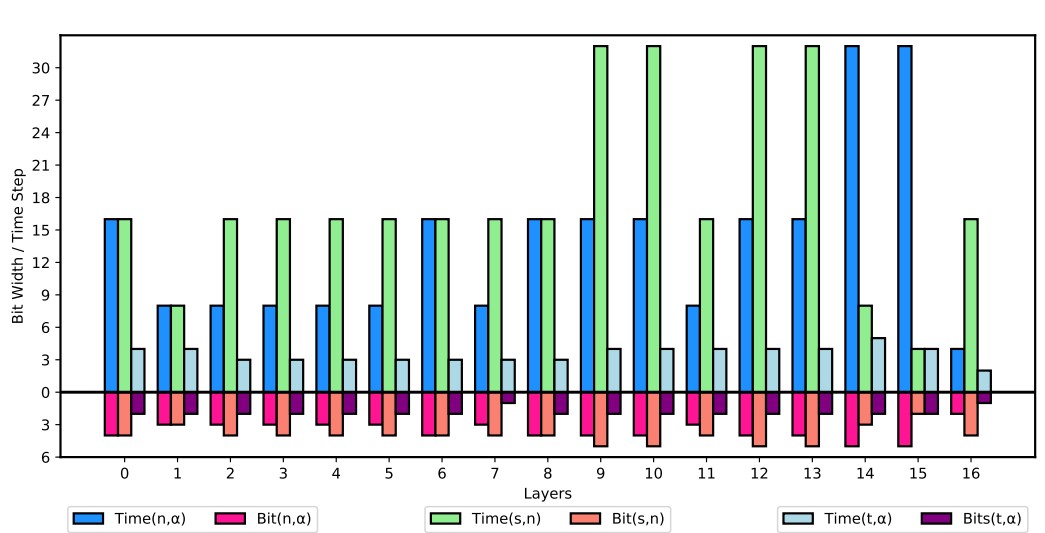

Figure 6: CIFAR-100, ResNet-18.

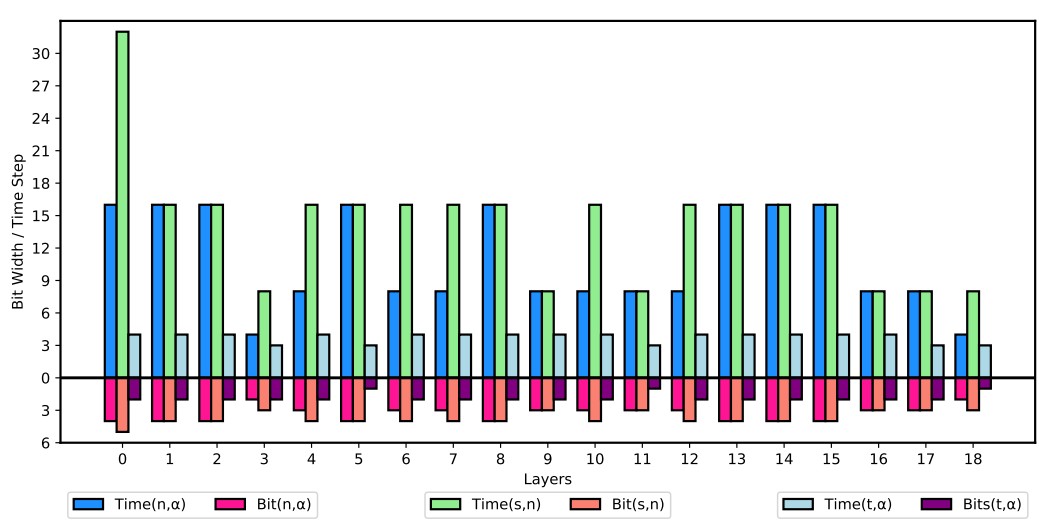

Figure 7: CIFAR-10, ResNet-20.

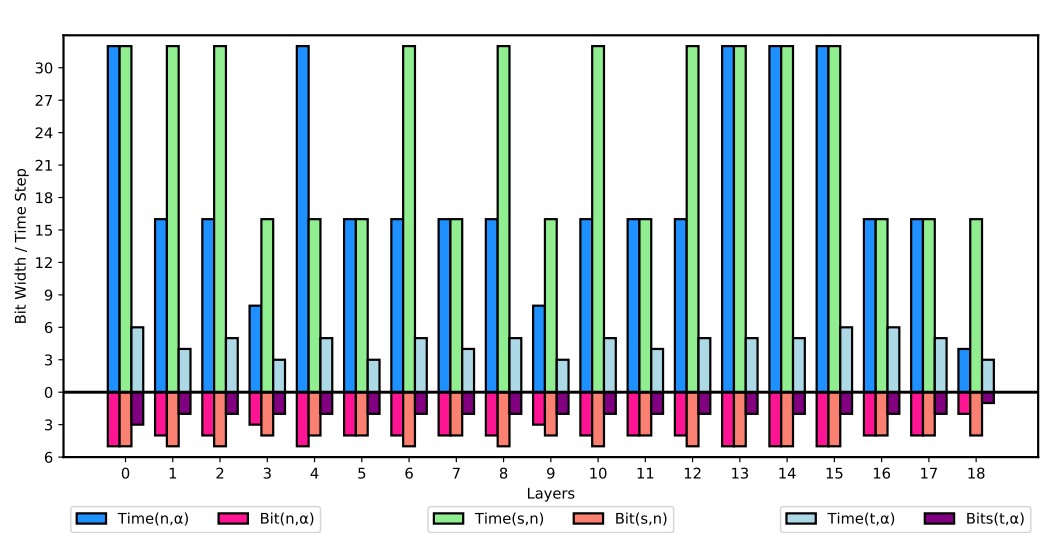

Figure 8: CIFAR-100, ResNet-20.

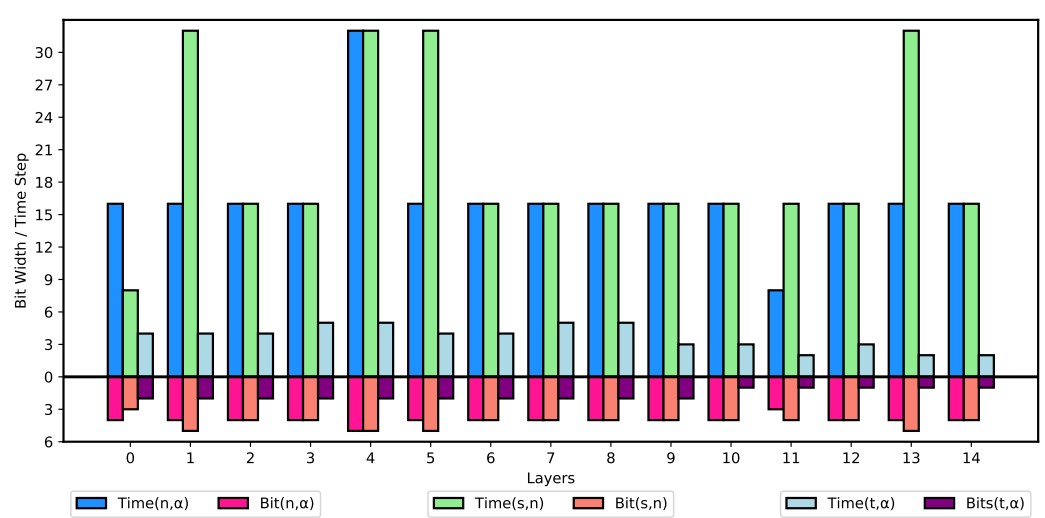

Figure 9: CIFAR-100, VGG-16.

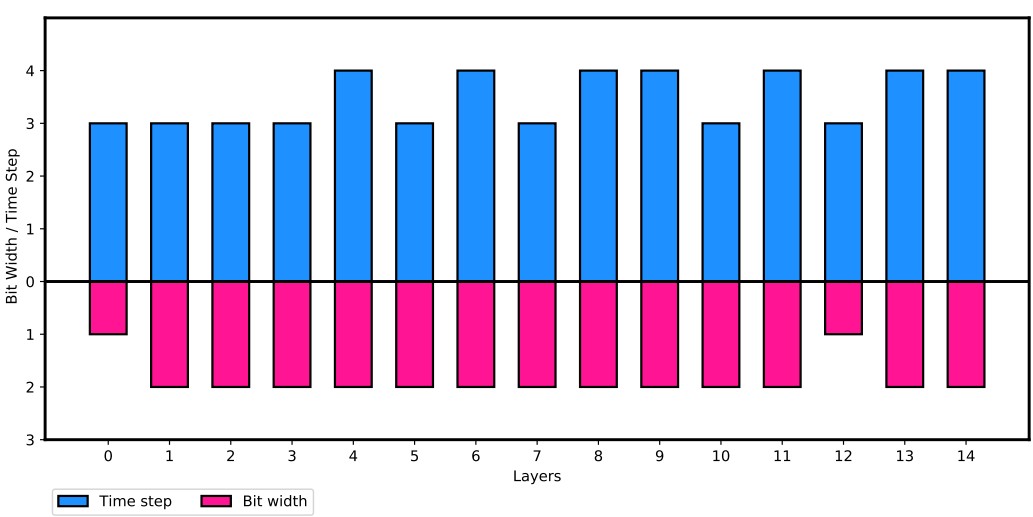

Figure 10: ImageNet, VGG-16.

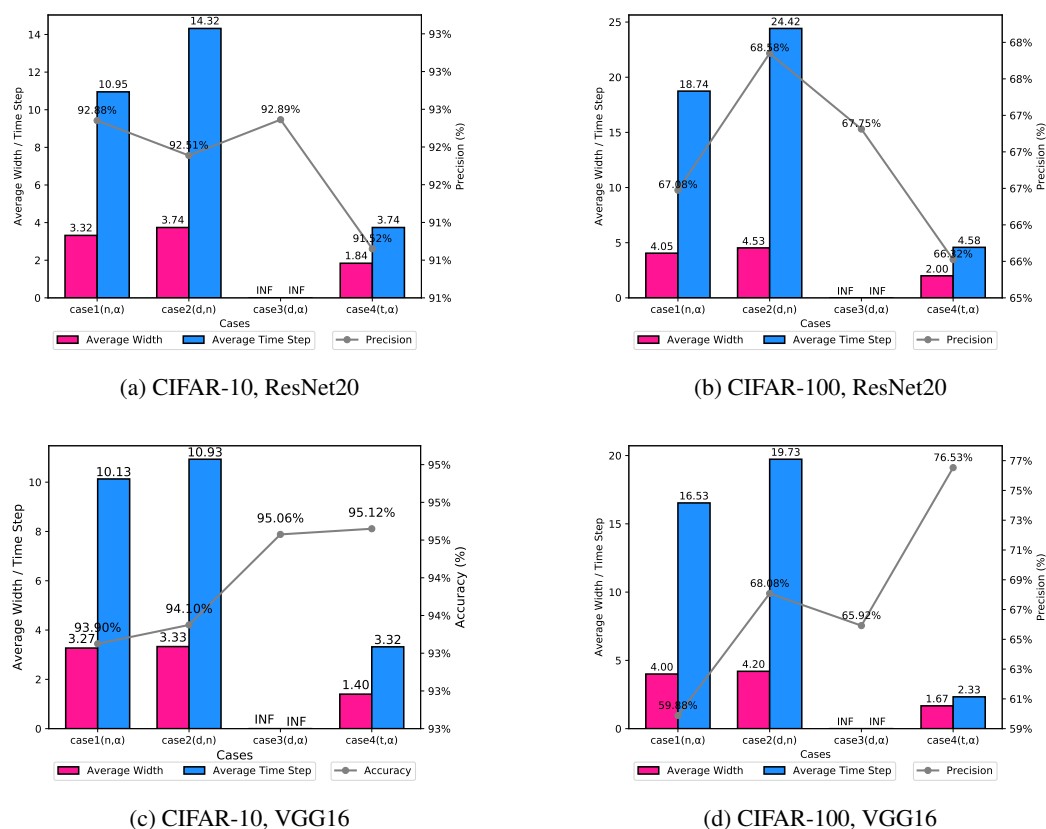

Figure 11: The average quantization bit-width, average time steps, and ANN accuracy under different quantization parameter schemes.