# OpenReview forum: "QAC:Quantization-Aware Conversion for Mixed-Timestep Spiking Neural Networks"
_ICLR.cc/2025/Conference — Submitted to ICLR 2025_

### Official Review · Reviewer_ejRD · 2024-10-27

**Soundness:** 2
**Presentation:** 2
**Contribution:** 2
**Rating:** 5
**Confidence:** 4

**Summary:**

This paper proposes Quantization-Aware Conversion (QAC) algorithm, which is a new ANN-SNN Conversion scheme based on mixed precision quantization and calibrating the membrane-related parameters (e.g. initial membrane potential and firing threshold).

**Strengths:**

1. In Eqs.(7)-(10) and Fig.2, the authors' analysis for four cases regarding parameter settings is interesting.
2. The authors calibrate the initial membrane potential and threshold under the condition of freezing pretrained weights.

**Weaknesses:**

1. In Fig.1, it seems that the inference time-steps of the converted SNN in different layers are various. Therefore, I would like to know if the authors choose vanilla IF model that emits binary spikes during the inference phase? Assume that the $l$-th layer adopts $T_l$ time-steps, when $T_l\neq T_{l+1}$, how do the IF neurons in the $l+1$-th layer receive the spike seqence $[\boldsymbol{s}^l(1),...,\boldsymbol{s}^l(T_l)]$ from the previous layer?

**Questions:**

See Weakness Section.

---

> ### Author Response · Authors · 2024-11-23
>
> > w1:In Fig.1, it seems that the inference time-steps of the converted SNN in different layers are various. Therefore, I would like to know if the authors choose vanilla IF model that emits binary spikes during the inference phase? Assume that the
> -th layer adopts
>  time-steps, when
> , how do the IF neurons in the
> -th layer receive the spike sequence
>  from the previous layer?
>
> Thank you for pointing out this important question. We would like to address your concerns and your questions in the following.
>
> **Neuron Model**:
> Yes, we use the vanilla Integrate-and-Fire (IF) neuron model in our SNN conversion framework. The IF neuron fires binary spikes during the inference phase and has been widely adopted due to its simplicity and hardware-friendly implementation.
>
> **Data Transmission Between Layers**:
> In mixed-timestep SNNs, the timestep discrepancies between layers are addressed through **Temporal Alignment** operations. A detailed explanation of this process is provided in Section 4.3 of the revised manuscript, along with ablation experiments (Table 4) that demonstrate the effectiveness of our temporal alignment methods.
>
> The final temporal alignment mechanism we adopted can be summarized as follows. Assuming the timestep of the $l-th$ layer is $T_l$, and the timestep of the $(l+1)-th$ layer is $T_{l+1}$,  $T_l \neq T_{l+1}$,  there are two cases:
>
> 1. Temporal Expansion: When $T_l < T_{l+1}$, the spike sequence from the $l-th$ layer is expanded to match $T_{l+1}$ by using  tempora expansion alignment techniques.
> 2. Temporal Reduction: When $T_l > T_{l+1}$, the spike sequence from the $l-th$ layer is reduced to $T_{l+1}$ by using  temporal reduction to ensure consistency with the next layer.
>
> To achieve temporal alignment for temporal expansion and reduction, we propose using the **temporal averaging expansion** method to adjust the input time dimensions. This method averages over the time dimension, and then replicates the result to match the desired time step length, which allows both temporal expansion and reduction to be performed using the same operation. we compute $ \bar{X}_l $ by averaging over the time dimension $T_l $, obtaining a tensor of shape $\bar{X}_l  \in \mathbb{R}^{B \times C_l \times H_l \times W_l}$:
>
> \begin{align}
> \bar{X}^l = \frac{1}{T_l} \sum_{t=1}^{T_l} X^l[t]
> \end{align}
>
> where, $ X^l{[t]}$ represents the input activation at time step $t$, and $ \bar{X}^l $ denotes the average result.
> Then, replicate $ \bar{X}^l $ along the time dimension to match the required timestep $ T_{l+1} $, expanding $\bar{X}^l$ to the required timesteps $T_{l+1} $, resulting in a tensor $ X_{l+1} \in \mathbb{R}^{T_{l+1} \times B \times C_l \times H_l \times W_l} $:
>
> \begin{align}
> X_{l+1} = \bar{X}^l \otimes 1_{T_{l+1}, 1}
> \end{align}
>
> where, $1_{T_{l+1}, 1}$ represents a tensor of length $T_{l+1} $, which replicates $\bar{X}^l $ across $ T_{l+1} $ time steps.
> Additionally, we also validated the **temporal averaging expansion**method on QCFS [1], where only a few lines of code were modified, resulting in an impressive accuracy improvement of over 10\%. The experimental results can be found in the appendix.
>
> [1] Tong Bu, Wei Fang, Jianhao Ding, PengLin Dai, Zhaofei Yu, and Tiejun Huang. Optimal ann-
> snn conversion for high-accuracy and ultra-low-latency spiking neural networks. arXiv preprint
> arXiv:2303.04347, 2023.

---

> > ### Author Response · Authors · 2024-11-23
> >
> > ### Comparison of QCFS Results with and without Temporal Averaging Expansion Alignment
> >
> > #### ResNet-20 on CIFAR-10
> >
> > | L    | Method         | T=1     | T=2     | T=3     | T=4     | T=8     | T=16    | T=32    | T=64    |
> > |------|----------------|---------|---------|---------|---------|---------|---------|---------|---------|
> > | L=2  | QCFS           | 78.85%  | 83.94%  | 86.43%  | 87.9%   | 89.69%  | 90.06%  | 89.97%  | 89.8%   |
> > |      | QCFS+(ave)     | 78.85%  | 88.58%  | 89.25%  | 89.31%  | 89.57%  | 88.96%  | 88.27%  | 89.77%  |
> > | L=4  | QCFS           | 62.32%  | 71.67%  | 78.21%  | 82.5%   | 89.48%  | 91.83%  | 92.5%   | 92.59%  |
> > |      | QCFS+(ave)     | 62.32%  | 87.30%  | 90.78%  | 91.90%  | 92.39%  | 92.54%  | 92.62%  | 92.61%  |
> > | L=8  | QCFS           | 52.68%  | 65.58%  | 73.48%  | 78.64%  | 88.31%  | 92.32%  | 93.21%  | 93.50%  |
> > |      | QCFS+(ave)     | 52.69%  | 83.47%  | 89.67%  | 91.54%  | 93.21%  | 93.54%  | 93.64%  | 93.70%  |
> > | L=16 | QCFS           | 36.45%  | 47.72%  | 56.95%  | 65.9%   | 84.12%  | 91.71%  | 93.22%  | 93.48%  |
> > |      | QCFS+(ave)     | 36.45%  | 75.29%  | 87.59%  | 91.13%  | 93.05%  | 93.59%  | 93.58%  | 93.57%  |
> >
> > #### VGG-16 on CIFAR-10
> >
> > | L    | Method         | T=1     | T=2     | T=3     | T=4     | T=8     | T=16    | T=32    | T=64    |
> > |------|----------------|---------|---------|---------|---------|---------|---------|---------|---------|
> > | L=2  | QCFS           | 61.45%  | 71.38%  | 74.57%  | 75.92%  | 77.38%  | 77.79%  | 77.79%  | 77.87%  |
> > |      | QCFS+(ave)     | 61.45%  | 77.61%  | 77.5%   | 77.91%  | 77.89%  | 77.98%  | 77.84%  | 77.88%  |
> > | L=4  | QCFS           | 10.89%  | 77.20%  | 84.35%  | 88.49%  | 93.33%  | 95.08%  | 95.76%  | 95.90%  |
> > |      | QCFS+(ave)     | 10.89%  | 91.02%  | 94.44%  | 95.33%  | 95.75%  | 95.94%  | 96.01%  | 95.99%  |
> > | L=8  | QCFS           | 72.03%  | 86.62%  | 92.03%  | 93.46%  | 95.07%  | 95.70%  | 95.74%  | 95.77%  |
> > |      | QCFS+(ave)     | 72.03%  | 93.32%  | 95.22%  | 95.64%  | 95.77%  | 95.83%  | 95.83%  | 95.84%  |
> > | L=16 | QCFS           | 29.02%  | 86.02%  | 89.38%  | 91.91%  | 94.65%  | 95.77%  | 96.03%  | 96.07%  |
> > |      | QCFS+(ave)     | 29.02%  | 92.84%  | 94.66%  | 95.39%  | 95.85%  | 96.04%  | 96.03%  | 96.04%  |
> >
> > #### ResNet-20 on CIFAR-100
> >
> > | L    | Method         | T=1     | T=2     | T=3     | T=4     | T=8     | T=16    | T=32    | T=64    |
> > |------|----------------|---------|---------|---------|---------|---------|---------|---------|---------|
> > | L=2  | QCFS           | 43.71%  | 44.68%  | 55.64%  | 58.17%  | 61.18%  | 62.09%  | 61.93%  | 61.56%  |
> > |      | QCFS+(ave)     | 43.71%  | 58.97%  | 60.34%  | 61.17%  | 60.92%  | 61.32%  | 61.14%  | 61.14%  |
> > | L=4  | QCFS           | 25.64%  | 36.00%  | 44.10%  | 50.36%  | 62.02%  | 66.33%  | 67.26%  | 67.05%  |
> > |      | QCFS+(ave)     | 25.64%  | 56.56%  | 63.66%  | 64.67%  | 66.11%  | 66.55%  | 66.76%  | 66.50%  |
> > | L=8  | QCFS           | 11.48%  | 17.11%  | 23.46%  | 29.94%  | 51.29%  | 64.65%  | 68.03%  | 68.62%  |
> > |      | QCFS+(ave)     | 11.48%  | 43.82%  | 59.06%  | 64.47%  | 67.96%  | 68.06%  | 68.52%  | 68.62%  |
> > | L=16 | QCFS           | 7.26%   | 11.15%  | 15.47%  | 20.54%  | 41.95%  | 61.81%  | 68.07%  | 69.02%  |
> > |      | QCFS+(ave)     | 7.26%   | 32.49%  | 53.15%  | 61.61%  | 68.28%  | 69.13%  | 69.23%  | 69.32%  |
> >
> > #### VGG-16 on CIFAR-100
> >
> > | L    | Method         | T=1     | T=2     | T=3     | T=4     | T=8     | T=16    | T=32    | T=64    |
> > |------|----------------|---------|---------|---------|---------|---------|---------|---------|---------|
> > | L=2  | QCFS           | 65.06%  | 68.97%  | 71.13%  | 72.3%   | 74.34%  | 75.13%  | 75.43%  | 75.60%  |
> > |      | QCFS+(ave)     | 65.06%  | 73.85%  | 74.34%  | 74.96%  | 75.56%  | 75.39%  | 75.54%  | 75.54%  |
> > | L=4  | QCFS           | 57.57%  | 64.33%  | 67.93%  | 70.13%  | 74.75%  | 76.33%  | 77.01%  | 77.15%  |
> > |      | QCFS+(ave)     | 57.57%  | 73.01%  | 75.53%  | 76.30%  | 76.90%  | 77.03%  | 77.08%  | 77.24%  |
> > | L=8  | QCFS           | 45.47%  | 55.55%  | 60.53%  | 64.93%  | 72.42%  | 76.02%  | 77.22%  | 77.44%  |
> > |      | QCFS+(ave)     | 45.47%  | 69.88%  | 74.58%  | 75.7%   | 75.58%  | 77.15%  | 77.14%  | 77.11%  |
> > | L=16 | QCFS           | 28.98%  | 41.11%  | 48.66%  | 54.41%  | 67.02%  | 74.39%  | 76.87%  | 77.56%  |
> > |      | QCFS+(ave)     | 28.98%  | 66.23%  | 73.58%  | 75.48%  | 76.86%  | 77.71%  | 77.68%  | 77.69%  |

---

### Official Review · Reviewer_6VHG · 2024-10-29

**Soundness:** 3
**Presentation:** 3
**Contribution:** 2
**Rating:** 6
**Confidence:** 4

**Summary:**

This paper presents an ANN-to-SNN conversion method, called Quantization-Aware Conversion (QAC). It reveals the relationship between the quantization levels of activations in ANNs and the timesteps in SNNs. Drawing inspiration from the mixed precision quantization technique used in ANNs, QAC allows for converting ANNs to SNNs with varying timesteps. Experimental results demonstrate that QAC achieves competitive performance compared to state-of-the-art methods with few timesteps.

**Strengths:**

1. This paper is technically solid. It details the equivalence between the quantization levels of activations in ANNs and the timesteps in SNNs, the four cases of gradient estimation, and the calibration of initial membrane potential.
2. The proposed QAC method yields SNNs with few timesteps.

**Weaknesses:**

1. The motivation for this technique is unclear. This paper reveals the equivalence between the quantization bit-width of ANN activations and the timesteps in SNNs with soft-threshold resets. However, if the quantized ANNs are equivalent to SNNs with soft reset, why don't we just use the quantized ANNs? I suggest the authors highlight the advantages of SNNs compared to the quantized ANNs.
2. The performance of the converted SNNs by the QAC method lags behind the state-of-the-art methods with more timesteps.

**Questions:**

Please refer to the weaknesses.

---

> ### Author Response · Authors · 2024-11-24
>
> > w1: The motivation for this technique is unclear. This paper reveals the equivalence between the quantization bit-width of ANN activations and the timesteps in SNNs with soft-threshold resets. However, if the quantized ANNs are equivalent to SNNs with soft reset, why don't we just use the quantized ANNs? I suggest the authors highlight the advantages of SNNs compared to the quantized ANNs.
>
> ### **1. I suggest the authors highlight the advantages of SNNs compared to the quantized ANNs**
> Thank you for raising this important question. While we acknowledge that the equivalence between quantized ANNs and SNNs is a key theoretical contribution of our work, the primary motivation for SNNs lies in their unique computational paradigm that offers significant energy efficiency, which quantized ANNs cannot match.
>
> Quantized ANNs, despite being efficient, still rely on fixed-point multiplications for their forward propagation. In contrast, SNNs use 0/1 spike activations, replacing multiplications with additions in hardware, which is inherently less energy-intensive.
> This efficiency stems from the fact that the energy cost of integer multiplication (INT8 MUL) is approximately 6.7× higher than integer addition (INT8 ADD), as supported by [1]. Therefore, SNNs offer a mechanism to transform multiplication-heavy operations into addition-based ones, making them highly advantageous in energy-constrained scenarios, such as edge computing and IoT devices.
>
> > The performance of the converted SNNs by the QAC method lags behind the state-of-the-art methods with more timesteps.
>
> ### **2. The performance of the converted SNNs by the QAC method lags behind the state-of-the-art methods with more timesteps.**
> We appreciate this constructive comment. The primary focus of QAC is to achieve competitive accuracy with as few timesteps as possible, optimizing latency and computational efficiency. While we acknowledge that some state-of-the-art methods may achieve higher accuracy under high-timestep settings, our study prioritizes performance trade-offs under low-timestep constraints.
> To address this concern, we conducted additional temporal scalability analyses on CIFAR-10 and CIFAR-100 datasets for ResNet-18, ResNet-20, and VGG-16, as shown in Figure 3 and Table 5. Key findings include:
> 1. Optimal Accuracy at 3T: Our method achieves the highest accuracy at three times the original timesteps (3T). For example:
>   - ResNet-18: On CIFAR-10, the accuracy reaches 95.99%, comparable to or surpassing the original ANN.
>   - ResNet-20: On CIFAR-100, the accuracy improves to 91.76%, significantly higher than with fewer timesteps.
>   - VGG-16: On CIFAR-10 and CIFAR-100, the accuracy trends are similar, achieving optimal performance at 3T.
> 2. Performance Saturation: Beyond 3T, the accuracy saturates and shows negligible improvement. This indicates that QAC efficiently utilizes time information under low-timestep conditions and demonstrates strong temporal scalability.
>
> To further strengthen our comparative analysis, we would like to kindly ask the reviewer to clarify which specific works you consider state-of-the-art (SOTA) for ANN-to-SNN conversion or direct SNN training. While we have compared our method with several recent approaches (e.g., SlipReLU, SGDND, QCFS), additional comparisons with your suggested SOTA methods can provide valuable insights and further contextualize our contributions.
>
> We will revise our paper to include this temporal scalability analysis and acknowledge limitations, while also incorporating comparisons with the SOTA methods highlighted by the reviewer if feasible.

---

> ### Comment · Reviewer_6VHG · 2024-11-27
>
> Thank you for your reply. My concerns have been addressed. However, I didn't find a corresponding revision in the paper. I would raise my rating if the authors could add comparisons with quantized ANNs to highlight the advantages of converted SNNs.

---

> ### Author Response · Authors · 2024-12-02
>
> Thank you  for your willingness to reconsider your rating of our paper. To address your concerns regarding the performance advantages of **Quantized ANNs (QANNs)** versus the converted **SNNs**, we conducted experiments using VGG-16, ResNet-18, and ResNet-20 on CIFAR-10 and CIFAR-100. The results demonstrate that **SNNs** achieve several times greater energy efficiency compared to **QANNs**, as shown in Table 2. Due to the PDF submission time constraints, only partial results were included in Appendix 7.8 of the submitted version. We will submit the full version of the paper on arXiv, which includes the complete experimental analysis and data.
>
> ## **Please find our detailed responses below.**
>
> In ANNs, the output activations are obtained by performing a multiplication and accumulation operation between the input activations and the weights. This operation is referred to as **multiply-accumulate operation (MACs)**. For example, $ x^l = W^l \cdot x^{l-1} $, where $ x^{l-1} $ is the output activation from the previous layer, and $ W^l $ is the weight matrix of the current layer. The elements of $ x^{l-1} $ and $ W^l $ are typically multi-bit fixed-point or floating-point values.
>
> In SNNs, the activations are represented by binary spikes, as a result, the **MACs** in SNN can be simplified to an **accumulation operation (ACs)** over the weights . For example, given an activation from  $l-1$-th layer $ x^{l-1} = [0, 1, 0, 1]^T $ and weights $ W^l = [w_1, w_2, w_3, w_4] $ in $l$-th layer, the output activation  of $l$-th layer is $ x^l = W^l \cdot x^{l-1} = w_2 + w_4 $. Therefore, SNNs can perform only addition operations which typically consume less energy and chip area than multiplication operations in hardware. This is because multiplication involves more complex computational logic and requires more hardware resources (e.g., multipliers), while addition only requires simple adder units. The specific energy consumption  [1] is shown in Table 1.
> ### **Table 1. Energy Table for 45nm CMOS Process**
>
> | **Operation**      | **Energy (pJ)** |
> |--------------------|-----------------|
> | 8b ADD (INT)       | 0.03            |
> | 32b ADD (INT)      | 0.1             |
> | 8b MULT (INT)      | 0.2             |
> | 32b MULT (INT)     | 3.1             |
> | 8b MAC (INT)       | 0.23            |
> | 16b ADD (FP)       | 0.4             |
> | 32b ADD (FP)       | 0.9             |
> | 16b MULT (FP)      | 1.1             |
> | 32b MULT (FP)      | 3.7             |
> | 1b-8b MULT (INT)*  | 0.025           |
> | 2b-8b MULT (INT)*  | 0.05            |
> | 3b-8b MULT (INT)*  | 0.075           |
>
> **Note: * means the data is derived from [2][3] [Nagendra et al., 1996](https://doi.org/10.1109/EDL.1996.535059) and [Potipireddi et al., 2013](https://doi.org/10.1109/TCAD.2012.2233991).**
>
> To compare the energy efficiency of **SNN** and **quantized ANN (QANN)**, we conducted experiments using VGG-16 , ResNet-18 and ResNet-20 on CIFAR-10 and CIFAR-100. We assume that the weights are all 8-bit fixed-point quantized (this assumption does not affect the final comparison, as both SNN and QANN use the same weights). The energy consumption of multiplication and addition operations  is provided in Table 1 [1]. Using this data, we can calculate the energy required for MACs. For example, for 8-bit fixed-point quantized activations and weights, the energy consumption of a multiply-accumulate operation is 0.23 pJ.
>
> ## **1. Energy Computation for Mixed-Width MACs.**
> However, because the activations of **QANN** in our work QAC are mixed-precision quantized, the **bit width of activations varies across different layers**, which results in multiply-accumulate operations **(MACs)** with different bit-width activations and weights, such as INT2 activations and INT8 weights, for which energy consumption values are not provided in [1] .
>
> To address this, we refer to the approach in [2][3], which states that the power consumption of an **adder** is **linearly** related to bitwidth[2], while the power consumption of a **multiplier** is **quadratic** with respect to bit width [3]. This allows us to compute the energy of **MACs** for different bit widths (as shown by the * data in Table 1). For instance, a 2-bit fixed-point multiplication (INT2) consumes 1/16 of the energy of an 8-bit fixed-point multiplication (INT8).
>
> Since SNN use binary spikes for activations, only accumulation operations **(ACs)** are required. Therefore, we can directly utilize the data from [1] without needing additional energy data from [2][3].

---

> ### Author Response · Authors · 2024-12-02
>
> ## **2. Energy consumption Calculation Method**
> We follow the energy calculation formula from [4] to compute the energy consumption of both the quantized ANN and the SNN, With $N$ input channels, $M$ output channels, weight kernel size $k \times k$, and output size $O \times O$,  the total FLOPS for ANN and SNN are as described in the equations below.
>
> $$
> FLOPS_{ANN} = O^2 \times N \times k^2 \times M \tag{1}
> $$
> $$
> FLOPS_{SNN} = O^2 \times N \times k^2 \times M \times f_r \tag{2}
> $$
>
>
> Where $f_r$ is the firing rate, representing the total number of firing neurons per layer, and usually $f_r\ll 1$ in SNNs.
>
> For energy calculation, each **MAC (for QANN)** or **AC (Addition operation, for SNN)** is considered, as below:
>
> $$
> E_{ANN} = \left( \sum_{l=1}^{N} FLOPS_{ANN} \right) \times E_{MAC}^{l}\tag{3}
> $$
> $$
> E_{SNN} = \left( \sum_{l=1}^{N} FLOPS_{SNN} \right) \times E_{AC} \times T_l \tag{4}
> $$
>
> Where $E_{MAC}^{l}$ represents the energy per MAC operation in l-th layer, **the $E_{MAC}^{l}$ vary across layers** due to the different quantization bit-widths of activations in each layer, $E_{AC}$ represents the energy consumed by addition operations, which is 0.03 pJ for INT8 quantized weights, **the $E_{AC}$ value is the same across all layers**. $ T_l $ denotes the time step of the $ l $-th layer in the SNN, and the **time steps may vary across different layers**.
>
>
> [1]. 1.1 computing’s energy problem (and what we can do about it). In 2014 IEEE international solid-state circuits conference digest of technical papers (ISSCC).
>
> [2]. Area-time-power tradeoffs in parallel adders. IEEE Transactions on Circuits and Systems II: Analog and Digital Signal Pro-
> cessing, 43(10):689–702, 1996.
>
>
> [3]. Automated hdl generation of two’s complement dadda multiplier with parallel prefix adders. Int J Adv Res Electr Electron Instrum Eng, 2, 2013.
>
>
> [4]. Toward scalable, efficient, and accurate deep spiking neural networks with backward residual connections, stochastic softmax, and hybridization. Frontiers in Neuroscience, 14:653, 2020.

---

> ### Author Response · Authors · 2024-12-02
>
> ## **3. Experimental Results and Data.**
> Table 2 presents a comparison of energy efficiency between **QANN** and **SNN** on CIFAR-10 and CIFAR-100,  as well as the ratio of energy consumption between **QANN** and **SNN** (denoted as $E_{QANN}$ / $E_{SNN}$ ) for various models.
>
> - For the **CIFAR-10** dataset, the energy consumption of **SNN** is significantly lower than that of  **QANN** across all models, with the **VGG-16** model having the highest ratio of energy efficiency (6.23x).
> - For the **CIFAR-100** dataset, similar trends are observed, with **SNN** being more energy-efficient than **QANN** and the ResNet-18 model showing a ratio of 4.94x.
>
>
> ### **Table 2. Comparison of Energy Efficiency Between Quantized ANN and SNN**
>
> | **Dataset**  | **Model**    | **Energy (mJ) QANN** | **Energy (mJ) SNN** | **$E_{QANN}$ / $E_{SNN}$** |
> |--------------|--------------|----------------------|---------------------|---------------------------|
> | **CIFAR-10** | VGG-16       | 0.0229               | 0.0037              | 6.23x                     |
> |              | ResNet-18    | 0.0409               | 0.0071              | 5.79x                     |
> |              | ResNet-20    | 0.0031               | 0.00098             | 3.15x                     |
> | **CIFAR-100**| VGG-16       | 0.0256               | 0.0051              | 5.03x                     |
> |              | ResNet-18    | 0.0428               | 0.0087              | 4.94x                     |
> |              | ResNet-20    | 0.0034               | 0.0012              | 2.75x                     |
>
> Tables 3 and 4 provide the necessary data to calculate the energy consumption of QANN and SNN, including the **activation quantization bit-widths** for different layers, the number of **time steps**, and the **spike firing rates**. Using these data, along with **Equations (3) and (4)**, the values in **Table 2** can be derived.
>
> ### **Table 3. ANN Activations Quantization Bit Width.**
>
> | **Dataset**  | **Model**    | **Activations Quantization Bit Width**                                                      |
> |--------------|--------------|---------------------------------------------------------------------------------------------|
> | **CIFAR-10** | VGG-16       | 2, 2, 2, 2, 2, 2, 1, 1, 1, 1, 1, 1, 1, 1, 1                                                 |
> |              | ResNet-18    | 2, 2, 2, 2, 2, 2, 2, 2, 2, 2, 2, 2, 2, 1, 1, 1, 1                                           |
> |              | ResNet-20    | 2, 2, 2, 2, 2, 1, 2, 2, 2, 2, 2, 1, 2, 2, 2, 2, 2, 2, 1                                     |
> | **CIFAR-100**| VGG-16       | 2, 2, 2, 2, 2, 2, 2, 2, 2, 2, 1, 1, 1, 1, 1                                                 |
> |              | ResNet-18    | 2, 2, 2, 2, 2, 2, 2, 1, 2, 2, 2, 2, 2, 2, 2, 2, 1                                           |
> |              | ResNet-20    | 3, 2, 2, 2, 2, 1, 2, 2, 2, 2, 2, 1, 2, 2, 3, 3, 2, 2, 2                                     |
> ### **Table 4.  SNN Time Steps and Firing Rate**
>
> | **Dataset**  | **Model**    | **Time Steps / Firing Rate**                                                                 |
> |--------------|--------------|----------------------------------------------------------------------------------------------|
> | **CIFAR-10** | VGG-16       | 3, 3, 3, 3, 3, 3, 3, 3, 2, 1, 1, 1, 2, 2, 2                                                |
> |              |              | 0.3 / 0.14 / 0.14 / 0.08 / 0.12 / 0.08 / 0.07 / 0.07 / 0.1 / 0.18 / 0.45 / 0.4 / 0.29 / 0.19 / 0.47 |
> |              | ResNet-18    | 3, 3, 3, 3, 3, 3, 3, 3, 3, 3, 3, 3, 3, 2, 2, 2, 2                                           |
> |              |              | 0.36 / 0.22 / 0.20 / 0.13 / 0.22 / 0.14 / 0.14 / 0.08 / 0.19 / 0.11 / 0.11 / 0.05 / 0.15 / 0.09 / 0.12 / 0.17 / 0.19 |
> |              | ResNet-20    | 4, 4, 4, 3, 4, 3, 4, 4, 4, 4, 4, 3, 4, 4, 4, 4, 4, 3, 3                                     |
> |              |              | 0.35 / 0.23 / 0.32 / 0.17 / 0.36 / 0.12 / 0.38 / 0.24 / 0.27 / 0.09 / 0.30 / 0.09 / 0.33 / 0.17 / 0.17 / 0.09 / 0.19 / 0.08 / 0.22 |
> | **CIFAR-100**| VGG-16       | 4, 4, 4, 5, 5, 5, 5, 5, 5, 3, 3, 2, 3, 2, 2                                                 |
> |              |              | 0.29 / 0.14 / 0.14 / 0.09 / 0.11 / 0.09 / 0.06 / 0.07 / 0.05 / 0.07 / 0.26 / 0.41 / 0.24 / 0.21 / 0.21 |
> |              | ResNet-18    | 4, 4, 3, 3, 3, 3, 3, 3, 3, 4, 4, 4, 4, 4, 5, 4, 2                                           |
> |              |              | 0.32 / 0.23 / 0.24 / 0.12 / 0.23 / 0.15 / 0.20 / 0.08 / 0.25 / 0.13 / 0.13 / 0.07 / 0.15 / 0.08 / 0.06 / 0.03 / 0.07 |
> |              | ResNet-20    | 6, 4, 5, 3, 5, 3, 5, 4, 5, 3, 5, 4, 5, 5, 5, 6, 6, 5, 3                                     |
> |              |              | 0.27 / 0.24 / 0.31 / 0.15 / 0.35 / 0.13 / 0.39 / 0.21 / 0.31 / 0.10 / 0.38 / 0.07 / 0.40 / 0.14 / 0.16 / 0.09 / 0.21 / 0.10 / 0.20 |

---

> > ### Author Response · Authors · 2024-12-03
> >
> > Dear Reviewer,
> >
> > As the rebuttal deadline is approaching , I would be extremely grateful if you could take a moment to review the changes and responses we have provided to your feedback. Your insights and suggestions have been invaluable, and we want to ensure that we’ve addressed all of your concerns to the best of our ability.
> >
> > If you have any remaining questions or comments, I would be happy to clarify or elaborate further. Thank you again for your time and effort in reviewing our paper, and I look forward to hearing from you.
> >
> > Best regards

---

> > > ### Comment · Reviewer_6VHG · 2024-12-03
> > >
> > > Thank you for your detailed reply. My concerns have been addressed. I have raised my rating.

---

> > > > ### Author Response · Authors · 2024-12-03
> > > >
> > > > Thank you for your thoughtful and constructive feedback. I greatly appreciate your time and effort in reviewing my submission. Your positive evaluation motivates me to continue improving my work.
> > > >
> > > > Wishing you all the best

---

### Official Review · Reviewer_cCZ2 · 2024-11-02

**Soundness:** 3
**Presentation:** 3
**Contribution:** 3
**Rating:** 6
**Confidence:** 3

**Summary:**

The paper presents a Quantization-Aware Conversion (QAC) method tailored for training Spiking Neural Networks (SNNs). A significant innovation highlighted is the establishment of a theoretical equivalence between the quantization bit-width of activations in Quantized Neural Networks (QNNs) and the timesteps used in SNNs. This equivalence forms the basis for a new training approach that optimizes both the computational efficiency and accuracy of SNNs. The paper substantiates these claims through experimental validation on standard datasets, including CIFAR-10, CIFAR-100, and ImageNet. Results indicate that the QAC method surpasses existing ANN-to-SNN conversion techniques in terms of both accuracy and operational efficiency

**Strengths:**

+ Introduction of a novel insight that links the quantization bit-width in ANNs to the timesteps in SNNs, supporting the use of mixed-timestep strategies for SNNs.

+ Development of a mixed-precision quantization-based conversion algorithm that translates ANNs into mixed-timestep SNNs, optimizing for both accuracy and computational efficiency.

+ Experimental results on various image recognition datasets, demonstrating improvements over traditional conversion methods in terms of both accuracy and timestep efficiency.

**Weaknesses:**

+ The training procedure for the Quantization-Aware Conversion (QAC) includes two distinct phases, necessitating an additional training step to calibrate the membrane potentials' initial values and thresholds. This essentially doubles the training cost. It is crucial that the paper assesses the impact of this calibration step both before and after its application. Including ablation studies to highlight the effects of this calibration would provide a clearer picture of its necessity and impact, which should be explicitly stated for a fair comparison with other methods.
+ The QAC method appears to employ a Quantization-Aware Training (QAT) approach to train a Quantized Neural Network (QNN) before converting it to an SNN, rather than supporting the direct conversion of existing ANNs into SNNs. This positions QAC more as a specialized training method rather than a straightforward conversion technique. To fairly evaluate the efficacy and practicality of QAC, comparative experiments against direct training methods should be conducted. These comparisons would effectively showcase the trade-offs involved, helping to better position QAC in the landscape of SNN training methodologies and underline its specific advantages and limitations.

**Questions:**

See Weaknesses.

---

> ### Author Response · Authors · 2024-11-23
>
> > w1: The training procedure for the Quantization-Aware Conversion (QAC) includes two distinct phases, necessitating an additional training step to calibrate the membrane potentials' initial values and thresholds. This essentially doubles the training cost. It is crucial that the paper assesses the impact of this calibration step both before and after its application. Including ablation studies to highlight the effects of this calibration would provide a clearer picture of its necessity and impact, which should be explicitly stated for a fair comparison with other methods.
>
> ### **1. Including ablation studies to highlight the effects of this calibration would provide a clearer picture of its necessity and impact, which should be explicitly stated for a fair comparison with other methods.**
>
> Thank you for your valuable suggestions regarding the training cost and the calibration step. We understand your concern about the additional training cost and agree that a clearer analysis and ablation study of the impact of the calibration step are necessary.
>
> First, we would like to clarify the necessity of the calibration step: One of the core innovations of the QAC method is the use of quantization-aware training (QAT) to generate quantized neural networks (QNNs), which are then converted into SNNs. The purpose of calibrating the membrane potential and thresholds is to optimize the initial membrane potential and thresholds, significantly reducing conversion error, especially in low-timestep scenarios, ensuring that the SNN achieves accuracy close to that of the QNN.
> - **Comparison from Ablation Studies:**
>  In the revised manuscript, we have included additional detailed ablation experiments to quantitatively evaluate the performance differences of SNNs before and after the calibration step. These ablation studies clarify the necessity of the calibration step for improving model performance and help reviewers and readers better understand the role of this additional step in the QAC method. Specifically, we conducted ablation experiments on ResNet18, ResNet20, and VGG-16 models across CIFAR-10, CIFAR-100, and ImageNet datasets. Detailed results can be found in Section 5.4 and Table 3 of the manuscript.
> ### Calibration Results: CIFAR-10 / CIFAR-100 / ImageNet
>
> | Calibration        | ResNet-18 (%)       | ResNet-20 (%)       | VGG-16 (%)         |
> |--------------------|---------------------|---------------------|--------------------|
> | QAC (Base)         | 95.31 / 76.56 / -  | 90.29 / 65.80 / -   | 94.45 / 76.24 / 63.21 |
> | QAC (Base+CAL)     | 95.29 / 77.81 / -  | 91.13 / 65.39 / -   | 94.78 / 76.37 / 66.38 |
>
> The results indicate that the calibration step (Base+CAL) improves accuracy by approximately 1% across datasets, with a 3% improvement on ImageNet. This demonstrates the effectiveness of the calibration step.
>
> ### **2. The training procedure for the Quantization-Aware Conversion (QAC) includes two distinct phases, necessitating an additional training step to calibrate the membrane potentials' initial values and thresholds. This essentially doubles the training cost. .**
>
> - **Analysis of the Additional Training Cost:**
> The calibration process is highly efficient. It does not rely on the original training dataset and is performed using the ANN outputs as targets during a lightweight optimization process lasting only a few dozen epochs. Therefore, the additional training cost is relatively low. Furthermore, we suggest that membrane potential and threshold calibration can be considered an optional component, to be applied based on specific needs in practical applications. Although the calibration step adds some training cost, we believe the resulting improvement in inference phase performance, particularly for applications requiring ultra-high accuracy, is worthwhile.
>
> > w2:The QAC method appears to employ a Quantization-Aware Training (QAT) approach to train a Quantized Neural Network (QNN) before converting it to an SNN, rather than supporting the direct conversion of existing ANNs into SNNs. This positions QAC more as a specialized training method rather than a straightforward conversion technique. To fairly evaluate the efficacy and practicality of QAC, comparative experiments against direct training methods should be conducted. These comparisons would effectively showcase the trade-offs involved, helping to better position QAC in the landscape of SNN training methodologies and underline its specific advantages and limitations.
>
> ### **1. To fairly evaluate the efficacy and practicality of QAC, comparative experiments against direct training methods should be conducted.**
>
> We appreciate your thoughtful considerations regarding the positioning of the QAC method. We understand your concerns about QAC being seen as a training method rather than a direct conversion technique, and we agree that comparisons with direct training methods are necessary to comprehensively evaluate the effectiveness of QAC.

---

> > ### Author Response · Authors · 2024-11-23
> >
> > - **Clarifying Positioning:**
> > First, we would like to clarify that the QAC method is not just a training approach but a framework for converting quantized neural networks (QNNs) into mixed-timestep SNNs. The goal of QAC is to enable SNNs to achieve efficient computational performance and high accuracy under low-timestep scenarios by introducing the quantitative relationship between quantization levels and timesteps. Although QAC employs QAT for QNN training, its **core contribution lies in the ANN-to-SNN conversion framework**, with the added capability of supporting SNNs with mixed timesteps.
> > - **Advantages Over Direct Training Methods:**
> > We have conducted experiments comparing the QAC method with traditional SNN direct training approaches (e.g., surrogate gradient-based methods) to demonstrate the differences in performance across various datasets. The results are as follows:
> > ### Table 8: Comparison with other directly training methods
> >
> > #### CIFAR-10
> >
> > | Method                   | Type               | Architecture        | Time-steps      | Accuracy (%)            |
> > |--------------------------|--------------------|---------------------|-----------------|-------------------------|
> > | tdBN (Zheng et al., 2021)   | Surrogate Gradient | ResNet-18           | 4               | 92.92                   |
> > | Dspike (Li et al., 2021b)   | Surrogate Gradient | ResNet-18           | 4               | 93.66                   |
> > | TET (Deng et al., 2022)     | Surrogate Gradient | ResNet-19           | 4               | 94.44                   |
> > | GLIF (Yao et al., 2022)     | Surrogate Gradient | ResNet-19           | 2, 4, 6         | 94.15, 94.67, 94.88     |
> > | RMP-Loss (Guo et al., 2023) | Surrogate Gradient | ResNet-20           | 20              | 91.89                   |
> > | PSN (Fang et al., 2024)     | Surrogate Gradient | Modified PLIF Net   | 4               | 95.32                   |
> > | TAB (Jiang et al., 2024a)   | Surrogate Gradient | VGG-9               | 4               | 93.41                   |
> > | TAB                        | Surrogate Gradient | ResNet-19           | 2, 4, 6         | 94.73, 94.76, 94.81     |
> > | **QAC (Ours)**             | ANN-to-SNN         | ResNet-18           | 2.76            | 95.29                   |
> > |                           | ANN-to-SNN         | ResNet-20           | 3.74            | 91.13                   |
> > |                           | ANN-to-SNN         | VGG-16              | 2.33            | 94.78                   |
> >
> > ---
> >
> > #### CIFAR-100
> >
> > | Method                   | Type               | Architecture        | Time-steps      | Accuracy (%)            |
> > |--------------------------|--------------------|---------------------|-----------------|-------------------------|
> > | Dspike (Li et al., 2021b)   | Surrogate Gradient | ResNet-18           | 4               | 73.35                   |
> > | TET (Deng et al., 2022)     | Surrogate Gradient | ResNet-19           | 4               | 74.47                   |
> > | GLIF (Yao et al., 2022)     | Surrogate Gradient | ResNet-19           | 2, 4, 6         | 75.48, 77.05, 77.35     |
> > | RMP-Loss (Guo et al., 2023) | Surrogate Gradient | ResNet-19           | 2, 4            | 74.66, 78.28, 78.98     |
> > | TAB (Jiang et al., 2024a)   | Surrogate Gradient | VGG-9               | 4               | 75.89                   |
> > | **QAC (Ours)**             | ANN-to-SNN         | ResNet-20           | 4.58            | 65.39                   |
> > |                           | ANN-to-SNN         | VGG-16              | 3.67            | 76.37                   |
> >
> > ---
> >
> > #### ImageNet-1k
> >
> > | Method                   | Type               | Architecture        | Time-steps      | Accuracy (%)            |
> > |--------------------------|--------------------|---------------------|-----------------|-------------------------|
> > | tdBN (Zheng et al., 2021)   | Surrogate Gradient | SEW ResNet-34       | 6               | 63.72                   |
> > | SEW ResNet (Fang et al., 2021)| Surrogate Gradient| SEW ResNet-34       | 6               | 67.04                   |
> > | TET (Deng et al., 2022)     | Surrogate Gradient | SEW ResNet-34       | 6               | 68.00                   |
> > | GLIF (Yao et al., 2022)     | Surrogate Gradient | SEW ResNet-34       | 6               | 67.52                   |
> > | TAB (Jiang et al., 2024a)   | Surrogate Gradient | SEW ResNet-34       | 4, 24           | 65.94, 67.78            |
> > | PSN (Fang et al., 2024)     | Surrogate Gradient | SEW ResNet-34       | 6               | 70.54                   |
> > | **QAC (Ours)**             | ANN-to-SNN         | VGG-16              | 3.47            | 66.38                   |

---

> > > ### Author Response · Authors · 2024-11-23
> > >
> > > - **From the perspective of accuracy and timestep efficiency**, our conversion framework shows either superior performance or comparable performance to directly trained SNNs using surrogate gradient methods.
> > >   - **From the perspective of training cost**, direct training methods require significant computational and memory overhead due to the temporal dimension in the training process. Forward propagation, backpropagation, and gradient calculations during direct training have memory requirements that are at least $O(T)$ times higher than those for training quantized ANNs (QAT), along with substantial computational costs. In contrast, the first step of the QAC method, which uses QAT, significantly reduces memory and computation costs, greatly shortening the training time.
> > >
> > > In the revised manuscript, we have included these experimental results in the appendix, along with a comprehensive analysis of the QAC method’s strengths and weaknesses in terms of accuracy, timestep efficiency, and training cost. This additional data aims to help readers better understand the specific advantages of QAC compared to other training and conversion methods.
> > >
> > > ---
> > > We believe these analyses clearly explain the positioning of our method and its advantages over direct training approaches, helping the reviewer understand the value of this design decision.

---

### Official Review · Reviewer_sfK2 · 2024-11-03

**Soundness:** 3
**Presentation:** 2
**Contribution:** 3
**Rating:** 6
**Confidence:** 5

**Summary:**

This paper proposes a novel Quantization-Aware Conversion (QAC) algorithm to optimize Spiking Neural Networks (SNNs). Based on the theorem that the quantization bit-width in ANN activations is equivalent to the timesteps in SNNs with a soft reset, the authors introduce a mixed-precision quantization-based conversion method. This approach enables ANNs to be efficiently converted into mixed-timestep SNNs, significantly reducing the required timesteps and improving accuracy during inference. Additionally, the paper presents a calibration method for the initial membrane potential and thresholds, further enhancing performance. Experimental results demonstrate that this method outperforms existing approaches on the several static image classification datasets.

**Strengths:**

The proposed method enables ANNs to be efficiently converted into mixed-timestep SNNs through mixed-precision quantization, reducing the average timesteps during inference while maintaining accuracy.

**Weaknesses:**

1. This work lacks sufficient innovation, as it primarily combines existing ANN mixed-precision quantization and ANN-SNN conversion methods to transform an ANN into a mixed-timestep SNN. Although the authors integrate these techniques, the approach remains largely unchanged, offering limited novelty. For example, their discussion of various gradient cases in QAT parameter selection closely aligns with previously referenced work [1]. Additionally, while the authors claim in their contributions to reveal that the ANN activation quantization bit-width is equivalent to the timestep in SNNs with a soft reset, similar insights have already been discussed in prior researches [2][3].
2. The authors convert an ANN into a mixed-timestep SNN, where different layers in the SNN may have different timesteps. However, they do not explain how data flows within such a mixed-timestep structure, how modules with different timesteps are connected, whether this can be implemented in hardware, and whether the asynchronous nature of the SNN can be maintained simultaneously.
3. In the experiments, the authors only tested on static datasets without using dynamic datasets. Could this be related to the question of asynchronous behavior mentioned previously?
4.Some figures in the paper take up considerable space while conveying limited information, and in some figures, the text is too small to read clearly. It is recommended to rearrange these figures for better clarity and efficiency of space.


References:
[1] UHLICH S, MAUCH L, CARDINAUX F, et al. Mixed Precision DNNs: All you need is a good parametrization[J]. arXiv: Learning,arXiv: Learning, 2019.
[2] You K, Xu Z, Nie C, et al. SpikeZIP-TF: Conversion is All You Need for Transformer-based SNN[J]. arXiv preprint arXiv:2406.03470, 2024.
[3] Shen G, Zhao D, Li T, et al. Are Conventional SNNs Really Efficient? A Perspective from Network Quantization[C]//Proceedings of the IEEE/CVF Conference on Computer Vision and Pattern Recognition. 2024: 27538-27547.

**Questions:**

Please see weaknesses.

---

> ### Author Response · Authors · 2024-11-23
>
> > w1:This work lacks sufficient innovation, as it primarily combines existing ANN mixed-precision quantization and ANN-SNN conversion methods to transform an ANN into a mixed-timestep SNN. Although the authors integrate these techniques, the approach remains largely unchanged, offering limited novelty. For example, their discussion of various gradient cases in QAT parameter selection closely aligns with previously referenced work [1]. Additionally, while the authors claim in their contributions to reveal that the ANN activation quantization bit-width is equivalent to the timestep in SNNs with a soft reset, similar insights have already been discussed in prior researches [2][3].
>
> Thank you for your thoughtful comments.  We would like to address your concerns and your questions in the following.
>
>
> ### **1. The discussion of various gradient cases in QAT parameter selection closely aligns with previously referenced work [1].**
>
> Thanks for your observation regarding the relationship between our work and reference [1]. Work [1] identify optimal parameterizations such as "U3" and "P3" for gradient descent in quantization optimization. While these insights inspired our use of gradient-based methods.
>
> Our work leverages the QAT parameterization strategy and gradient approximation techniques outlined in [1], but extends and adapts these ideas into to a new context, with distinct innovations and contributions. Specifically, our contribution lies in the theoretical quantification of timestep and bit width relationships and its implementation in mixed-timestep SNN conversion algorithms. The differences between our work and [1] are as follows:
>
> - **Focus on SNN Conversion**:
>  Reference [1] primarily investigates the impact of mixed-precision quantization on both activations and weights in ANNs, proposing a general QAT optimization framework. In contrast, our work focuses on using quantization-aware training specifically for ANN-to-SNN conversion, limiting quantization to activations only to better align with the sparse spike-based computation in SNNs.
>
> - **Introduction of a New Parameter Combination Scheme**:
>  We propose a novel parameterization scheme, [t,α] and demonstrate through theoretical and experimental analysis that it achieves superior performance in the SNN conversion context, resulting in higher accuracy and fewer timesteps compared to existing schemes, such as [n, d], explored in [1].
>
> ### **2. The ANN activation quantization bit-width is equivalent to the timestep in SNNs with a soft reset, similar insights have already been discussed in prior researches [2][3].**
>
> Thank you for referencing papers [2, 3] and acknowledge their valuable contributions to the study of neural network quantization and SNN efficiency. Below, we will outline the distinctions between these works and ours, highlighting the unique contributions and significance of our research.
>
> #### **2.1 Discussion About work[3]**
> - Work [3] proposed a “Bit Budget” framework to analyze resource allocation between weight and activation quantization bit-widths and SNN timesteps. This framework focuses on **empirical analyses**, primarily exploring the energy-efficiency and accuracy trade-offs under different quantization conditions , but does not propose specific methods for timestep optimization..
> - Our Distinction:
> 1. We **quantified the specific relationship** between ANN quantization bit-width and SNN timesteps for the first time, expressed as **$T_{SNN} = 2^n_{ANN}$**. This formula provides a solid theoretical foundation for timestep optimization in SNN design, significantly extending the qualitative insights in [3] and providing a quantitative foundation for timestep optimization in ANN-to-SNN conversion.
> 2. It also directly guides the design of mixed-timestep SNNs. We developed a Quantization-Aware Conversion (QAC) algorithm based on our theoretical findings, which enables **layer-wise timestep optimization**. This approach reduces the required timesteps for inference while maintaining or even improving model accuracy. Our hierarchical optimization strategy, grounded in theory and practice, is a novel contribution not previously addressed in other work.
> 3. We validated our contributions through extensive experiments, demonstrating a superior balance between accuracy and efficiency.
>
> #### **2.2 Discussion About work[2]**
> - Work [2] focuses on ANN-to-SNN conversion, specifically for Transformer architectures. Work [2] identifies a specific timestep, referred to as $T_{up}$, beyond which the model's accuracy improves sharply. Notably, as shown in Figure 5(c)(d) of [2], $T_{up}$ is proportional to the quantization level. However, similar to word [3], work [2] only provides an **empirical relationship**. While it indicates a proportional relationship between $T_{up}$ and the quantization level, it does not establish a quantified model or provide further theoretical support.

---

> ### Author Response · Authors · 2024-11-23
>
> Compared to the above works [1, 2, 3], our study introduces a more comprehensive and systematic theoretical framework.
> 1. For the first time, we explicitly establish the **direct mathematical relationship** between ANN activation quantization bit-width and SNN timesteps, and we quantify this relationship in Theorem 1: $T = 2^n$, where n represents the bit-width of activation quantization, and T denotes the SNN timesteps. This explicit quantitative relationship has not been explored in References [2], [3], or other related works. Through this theoretical framework, we provide a solid foundation for timestep optimization based on quantization, laying the groundwork for future SNN conversion method development.
>
> 2.  By connecting mixed-precision quantization with our theoretical insights, we demonstrate that mixed-precision quantization not only enhances efficiency during the training phase but also enables timestep optimization during the SNN inference phase. This **layer-wise timestep optimization** approach is introduced for the first time in the literature.
>
> > w2:The authors convert an ANN into a mixed-timestep SNN, where different layers in the SNN may have different timesteps. However, they do not explain how data flows within such a mixed-timestep structure, how modules with different timesteps are connected, whether this can be implemented in hardware, and whether the asynchronous nature of the SNN can be maintained simultaneously.
> Thank you for pointing out this important question. We would like to address your concerns and your questions in the following.
>
>
> ### **1.  How data flows within such a mixed-timestep structure? How modules with different timesteps are connected?**:
>
> In mixed-timestep SNNs, the timestep discrepancies between layers are addressed through **Temporal Alignment** operations. A detailed explanation of this process is provided in Section 4.3 of the revised manuscript, along with ablation experiments (Table 4) that demonstrate the effectiveness of our temporal alignment methods.
>
> The final temporal alignment mechanism we adopted can be summarized as follows. Assuming the timestep of the $l-th$ layer is $T_l$, and the timestep of the $(l+1)-th$ layer is $T_{l+1}$,  $T_l \neq T_{l+1}$,  there are two cases:
>
> 1. Temporal Expansion: When $T_l < T_{l+1}$, the spike sequence from the $l-th$ layer is expanded to match $T_{l+1}$ by using  tempora expansion alignment techniques.
> 2. Temporal Reduction: When $T_l > T_{l+1}$, the spike sequence from the $l-th$ layer is reduced to $T_{l+1}$ by using  temporal reduction to ensure consistency with the next layer.
>
> To achieve temporal alignment for temporal expansion and reduction, we propose using the **temporal averaging expansion** method to adjust the input time dimensions. This method averages over the time dimension, and then replicates the result to match the desired time step length, which allows both temporal expansion and reduction to be performed using the same operation. we compute $ \bar{X}_l $ by averaging over the time dimension $T_l $, obtaining a tensor of shape $\bar{X}_l  \in \mathbb{R}^{B \times C_l \times H_l \times W_l}$:
>
> \begin{align}
> \bar{X}^l = \frac{1}{T_l} \sum_{t=1}^{T_l} X^l[t]
> \end{align}
>
> where, $ X^l{[t]}$ represents the input activation at time step $t$, and $ \bar{X}^l $ denotes the average result.
> Then, replicate $ \bar{X}^l $ along the time dimension to match the required timestep $ T_{l+1} $, expanding $\bar{X}^l$ to the required timesteps $T_{l+1} $, resulting in a tensor $ X_{l+1} \in \mathbb{R}^{T_{l+1} \times B \times C_l \times H_l \times W_l} $:
>
> \begin{align}
> X_{l+1} = \bar{X}^l \otimes 1_{T_{l+1}, 1}
> \end{align}
>
> where, $1_{T_{l+1}, 1}$ represents a tensor of length $T_{l+1} $, which replicates $\bar{X}^l $ across $ T_{l+1} $ time steps.
> Additionally, we also validated the **temporal averaging expansion**method on QCFS [1], where only a few lines of code were modified, resulting in an impressive accuracy improvement of over 10\%. The experimental results can be found in the appendix F7.6.
>
> [1] Tong Bu, Wei Fang, Jianhao Ding, PengLin Dai, Zhaofei Yu, and Tiejun Huang. Optimal ann-
> snn conversion for high-accuracy and ultra-low-latency spiking neural networks. arXiv preprint
> arXiv:2303.04347, 2023.

---

> > ### Comment · Reviewer_sfK2 · 2024-11-28
> >
> > Thank you for your detailed response and the updates to the manuscript. I appreciate the effort you have made to address the concerns raised in the initial review. However, while the temporal averaging expansion method demonstrates the best performance, I am concerned that averaging across the temporal dimension could convert spike data into fractional values, potentially undermining the inherent spiking nature of the SNN.

---

> > > ### Author Response · Authors · 2024-11-28
> > >
> > > Thank you for your feedback and concern regarding the averaging operation. Your question is very valuable, and here is my response to clarify the issue: The averaging operation does not convert spikes into floating-point values, and it does not disrupt the discrete spike nature of SNNs.
> > >
> > > This is because the averaging alignment operation is not performed after the IF activation layer, but rather after the convolutional or fully connected layer and before the IF activation function. Therefore, the spike information is still passed between adjacent layers.
> > >
> > > To better explain this, let’s consider a two-layer network as an example.
> > >
> > > For a vanilla SNN, the inference process is:
> > > **Conv1|IF1 → Conv2|IF2**
> > > In this case, the timesteps of the IF neurons in adjacent layers 1 and 2 are the same $T_1 = T_2$ and information is passed between the layers through spike activations.
> > >
> > > In QAC, the averaging alignment operation occurs before the IF layer:
> > > **Conv1|aver1|IF1 → Conv2|aver2|IF2**
> > > Although $T_1 \neq T_2$, meaning that the output from **IF1** in the first layer, $[T_1, B_1, C_1, H_1, W_1]$, does not match the expected timestep $T_2$ in **IF2**  of the second layer, the averaging operation **aver2** aligns $T_1$ to $T_2$ before **IF2** . This ensures that the information is still passed in the form of spikes between the two adjacent layers.
> > >
> > > We will open-source the code. The code will make it easy to understand and verify this process.

---

> > > > ### Comment · Reviewer_sfK2 · 2024-12-02
> > > >
> > > > Thank you for your response and revisions. My primary concerns have been addressed, so I am raising my score.

---

> > > > > ### Author Response · Authors · 2024-12-02
> > > > >
> > > > > We would like to sincerely thank the reviewer for reconsidering their rating of our paper. Your comments have significantly helped us improve the quality of our work. We truly appreciate the time and effort you spent in reviewing our work.

---

> ### Author Response · Authors · 2024-11-23
>
> ### **2.  Whether this can be implemented in hardware? and whether the asynchronous nature of the SNN can be maintained simultaneously?**:
>
> Thanks for your comments. The QAC method can be implemented on SNN accelerator while maintaining the asynchronous properties of SNNs. The detailed discussion can be found in the appendix F7.7. There are two mainstream implementations of SNN hardware: ANN accelerator variants and non-Von Neumann decentralized multi-core architectures (e.g., TrueNorth [1], Loihi [2]) [3].
>
> 1. The ANN accelerator variants [4] primarily achieve asynchronous computation by sending non-zero inputs to processing element (PE) arrays and performing spike matrix computations. These accelerators compute only a portion of the neural network at a time and process the entire network through iterative loops. The averaging alignment operations do not alter the original data flow, enabling such SNNs to run on this type of hardware.
>
> 2. On the other hand, multi-core brain-inspired hardware requires deploying the neurons of all layers across different cores. Neurons execute spike-based computations immediately upon receiving spike events, thereby achieving asynchronous execution. The network operates on the hardware in a pipelined manner. Although mixed-timestep SNNs can operate on such hardware, pipeline stalls may occur, introducing computational delays and potentially preventing the hardware from achieving optimal performance.
>
> [1] Loihi: A neuromorphic manycore processor with on-chip learning.
>
> [2] TrueNorth: Design and Tool Flow of a 65 mW 1 Million Neuron Programmable Neurosynaptic Chip.
>
> [3] Brain-Inspired Computing: A Systematic Survey and Future Trends.
>
> [4] FireFly v2: Advancing Hardware Support for High-Performance Spiking Neural Network With a Spatiotemporal FPGA Accelerator.
>
> >w3: In the experiments, the authors only tested on static datasets without using dynamic datasets. Could this be related to the question of asynchronous behavior mentioned previously? 4.Some figures in the paper take up considerable space while conveying limited information, and in some figures, the text is too small to read clearly. It is recommended to rearrange these figures for better clarity and efficiency of space.
>
> ### **1.  The authors only tested on static datasets without using dynamic datasets. Could this be related to the question of asynchronous behavior mentioned previously?**:
>
> Thanks for your comments. We agree that dynamic datasets could more comprehensively validate the asynchronous characteristics of SNNs. However, the primary objective of this paper is to demonstrate the performance of the QAC method in standard ANN-to-SNN conversion scenarios, particularly focusing on achieving low timesteps and high accuracy in static image classification tasks.
>
> To the best of our knowledge, most of current ANN-SNN conversion frameworks (e.g., [1, 2, 3, 4, 5, 6, 8, 9, 10]) predominantly address classification tasks on static datasets. This limitation arises because ANN-SNN conversion aims to utilize pretrained weights from ANNs, which are typically trained on static datasets. Consequently, the conversion framework processes static datasets only, independent of asynchronous behavior.
>
> In future work, we plan to extend the QAC method to dynamic datasets (e.g., DVS128-Gesture), thereby further verifying its applicability to temporal tasks. This extension will allow us to explore the method's potential for broader applications, including those involving dynamic or spatiotemporal data.
>
> [1] Optimal ANN-SNN Conversion for High-accuracy and Ultra-low-latency Spiking Neural Networks
>
> [2] Optimal ANN-SNN Conversion for Fast and Accurate Inference in Deep Spiking Neural Networks
>
> [3] Optimal Conversion of Conventional Artificial Neural Networks to Spiking Neural Networks
>
> [4] Optimized Potential Initialization for Low-Latency Spiking Neural Networks
>
> [5] A Unified Optimization Framework of ANN-SNN Conversion: Towards Optimal  Mapping from Activation Values to Firing Rates
>
> [6] Reducing ANN-SNN Conversion Error through Residual
>
> [7] A Free Lunch From ANN: Towards Efﬁcient, Accurate Spiking Neural Networks Calibration
>
> [8] Error-Aware Conversion from ANN to SNN via Post-training Parameter Calibration
>
> [9] SPATIO-TEMPORAL APPROXIMATION: A TRAINING- FREE SNN CONVERSION FOR TRANSFORMERS
>
> [10]  Towards High-performance Spiking Transformers from ANN to SNN Conversion
>
> ### **2. It is recommended to rearrange these figures for better clarity and efficiency of space.**:
> Thank you for your suggestions . In the revised manuscript, we have made the following optimizations:
>
> 1. Rearranged the layout of the figures, compressing those that occupied excessive space. For example, Figure 2 was reduced from four images to two.
>
> 2. Increased the font size of all figure text to ensure clarity and readability.
>
> 3. Merged closely related figures in Figure 3 to improve space utilization and redrew Figure 3 accordingly.

---

### Meta-Review · Area_Chair_XX3g · 2024-12-22

**Metareview:**

This paper combines Quantization-Aware Conversion (QAC) algorithm with membrane-related parameter calibration to achieve the inference of converted SNNs on mixed time-steps. Reviewers acknowledge the authors' detailed analysis in establishing the connection between quantization ANN and SNN, but express concerns about the specific calculation process and cost of the mixed-timestep network structure in the inference stage. After the discussion period, three reviewers rate 6 and one reviewer rates 5. None of the reviewers strongly argue for acceptance. Therefore, the final decision is not to accept the paper.

**Additional Comments On Reviewer Discussion:**

Reviewer sfK2 and ejRD express concern about how mixed time-steps are connected between different layers of the converted SNNs, while Reviewer cCZ2 and 6VHG argue about the fairness and performance of this work in terms of accuracy comparison. The authors conduct extensive supplementary discussion and experiments during the response phase. Overall, this paper would benefit from further clarification and analysis about the inference performance and computational overhead of QAC. In addition, I notice that some results reported by the authors in Table 8, ImagNet-1k seem to be inconsistent with the results in the original text (e.g. the inference time-step of SEW, TET, GLIF, TAB, PSN in their original text is 4 rather than 6).

---

### Decision · Program_Chairs · 2025-01-22

Reject